# UNIFIED LONG-TERM TIME-SERIES FORECASTING BENCHMARK

## ABSTRACT

In order to support the advancement of machine learning methods for predicting time-series data, we present a comprehensive dataset designed explicitly for long-term time-series forecasting. We incorporate a collection of datasets obtained from diverse, dynamic systems and real-life records. Each dataset is standardized by dividing it into training and test trajectories with predetermined lookback lengths. We include trajectories of length up to 2000 to ensure a reliable evaluation of long-term forecasting capabilities. To determine the most effective model in diverse scenarios, we conduct an extensive benchmarking analysis using classical and state-of-the-art models, namely LSTM, DeepAR, NLinear, N-Hits, PatchTST, and LatentODE. Our findings reveal intriguing performance comparisons among these models, highlighting the dataset-dependent nature of model effectiveness. Notably, we introduce a custom latent NLinear model and enhance DeepAR with a curriculum learning phase. Both consistently outperform their vanilla counterparts.

## 1 INTRODUCTION

Time-series (TS) forecasting is a fundamental task in contemporary data analysis, aiming to predict future events based on historical data. Its broad applicability across various domains, including strategic planning, resource allocation, risk management, and control design, underscores its importance. Consequently, extensive research efforts have been dedicated to developing efficient and accurate long-term time-series forecasting (LTSF) methods within the machine learning (ML) and applied statistics communities. Determining the optimal default method for a specific LTSF task and ensuring reliable out-of-the-box performance remains a challenge. The landscape of available methods encompasses neural networks (NN), continuous time models, statistical techniques, and more. Each method class has its advantages and an active research community. Notably, NN models have recently experienced rapid advancements in the context of the LTSF problem (Lim & Zohren, 2021; Torres et al., 2021; Lara-Benítez et al., 2021), motivating our focus on evaluating such models within our benchmark. In LTSF research, the availability of large and diverse datasets is crucial for training and benchmarking contemporary ML models. While natural language processing (NLP) benefits from abundant data, time-series datasets still lag behind in quantity and diversity. As highlighted by Zhou et al. (2023), the largest existing dataset for time-series analysis falls below 10GB, whereas our dataset exceeds 100 GB.

Existing LTSF datasets can be broadly classified into two categories. First, real-life datasets are sourced from diverse domains, such as temperature records, traffic data, and weather observations. However, these datasets often suffer from being univariate or are treated dimension by dimension, limiting their suitability for comprehensive evaluation. Second, researchers resort to custom synthetic datasets, including benchmarks like the long-range arena (Tay et al., 2021) for TS classification or data generated from chaotic dynamical systems (Gilpin, 2021). Nevertheless, relying solely on restricted scenario datasets hampers fair performance comparisons across state-of-the-art methods and impedes determining the best overall approach. Furthermore, NN trained by supervised learning is susceptible to overfitting, which may also be the case in LTSF, and some models may excel on specific benchmarks while struggling to generalize across different domains.

We introduce a unified LTSF benchmark dataset encompassing data from diverse domains to address the abovementioned limitations. We curate instances from established real-life datasets while leveraging synthetic data generation techniques to construct clean trajectories of arbitrary length

without missing values. By providing this unified dataset, we aim to enable fair and comprehensive evaluations of LTSF methods, promoting a deeper understanding of their strengths and weaknesses across various domains. In practical applications of LTSF, methods capable of producing predictions valid over extended periods are highly desirable. It is common practice to evaluate LTSF approaches using extended forecasted sequence lengths, reaching up to 720 time steps. However, we observe a lack of standardization regarding the length of the lookback window used as input for forecasting models. The lookback window is often fine-tuned for a specific model. For example, linear models are typically trained with long, fixed lookbacks, while transformer-based models employ shorter, variable-length histories. In this work, we propose a unified TS forecasting benchmark that addresses the identified gaps in existing benchmarks.

*1. Dataset Diversity:* We recognized dataset diversity's importance in comprehensively evaluating LTSF methods. To address this gap, we curated datasets from various domains, including robotic simulations with stochastic and deterministic control, partial differential equations, deterministic chaotic dynamics, and real-life data. By incorporating datasets from diverse domains, we aim to provide a more representative benchmark for LTSF.

*2. Facilitating Training and Testing ML Models:* We split the datasets into training and testing trajectories to ensure standardized evaluations and comparisons. Each separate trajectory has a fixed length, typically set to 1000 time steps and up to 2000. This approach allows for consistent and reproducible evaluations across different methods. We explicitly provide the desired history window lengths for each dataset, which provides a notion of various difficulty levels, e.g., shorter lookbacks make the forecasting task more challenging; on the other hand, longer lookbacks make encoding of the history more challenging. We emphasize that for the synthetic datasets, opposite to real-life datasets, each trajectory is separate and different, i.e., there is no overlap; hence, different lookback lengths define different problems.

*3. Introduction of New Hand-Crafted Models:* We recognized the need to explore and evaluate new models not tested before in the context of LTSF. Our research introduces two models: the latent NLinear model and DeepAR enhanced with curriculum learning (CL). These models have not been previously applied to LTSF tasks, and their inclusion in the benchmark is intended to highlight their performance and potential. We observed significant improvements across the entire dataset when using these models, suggesting their effectiveness as baselines for LTSF.

Having the unified dataset, we perform a thorough benchmark of a suite of NN-based methods, including classical approaches like LSTM, DeepAR, and latent ordinary differential equation (ODE) with recently published newer approaches demonstrated to improve the forecasting accuracy over the existing methods: SpaceTime, N-Hits, LTSF NLinear, and Patch transformer. Lastly, we share an open-source library with the implementation to help accelerate progress in the field.

## 1.1 Related Work

The most dominant class of LTSF benchmarks relies on datasets with rather uniform characteristics composed of nine widely-used real-world datasets: Electricity Transformer Temperature (ETT) from Zhou et al. (2021a) (split into four cases: ETTh1, ETTh2, ETTm1, ETTm2), Traffic, Electricity, Weather, ILI, ExchangeRate. All of them are univariate time series with a significant degree of non-deterministicity (these are real-life measurements). Our unified LTSF benchmark dataset consists of a few distinct classes of LTSF datasets mixing uni/multi-variate and real-life/synthetic. The long-range arena benchmark by Tay et al. (2021) introduces synthetic binary images used as a time-series classification challenge. In the NLP domain, synthetically generated data are increasingly used for training models capable of basic reasoning at smaller scales (Gunasekar et al., 2023; Li et al., 2023; Eldan & Li, 2023). In this work, we introduce a few synthetic TS datasets and focus exclusively on the forecasting task. A separate set of benchmarks with irregularly sampled time-stamps exist but is not suitable for long-term forecasting, the challenge being in irregular sample modeling: Electronic Health Records (Physionet) the data set of the Physionet Computing in Cardiology Challenge 2012 (Silva et al., 2012) and Climate Data (USHCN) The United States Historical Climatology Network (USHCN) dataset. The Physionet dataset contains data of 8000 ICU patients and 72 different time points per patient on average. Monash Time Series Forecasting Archive (Godahewa et al., 2021) compiles a set of 30 univariate real-life datasets from various domains. It evaluates statistical approaches for LTSF, Bauer et al. (2021) introduces a larger univariate real-life time-series collection.

In this work, we benchmark NN-based approaches using diverse TS datasets: uni/multi-variate and real-life/synthetic. Another collection of uniform characteristics chaotic datasets dedicated for TS forecasting by Gilpin (2021) benchmark various ML models in this uniform setting. The UCR time series archive is dedicated to time-series classification task (Dau et al., 2019). Refer to Tab. 1 for a summary of related benchmark datasets. It is visible that our compressed dataset is significantly larger than the other available TS benchmark datasets; we choose the compressed size metric for comparison, as it reflects trajectory overlaps; in particular, real-life datasets are more amenable for compression than synthetic ones. For instance, 20k synthetic trajectories with up to 2k timestamped states result in 40M distinct states in total.

Table 1: Comparison of our benchmark dataset to prior work.

| Benchmark | compressed size | origin | domain | forecasting | classification | eval. methods |
|---|---|---|---|---|---|---|
| ETT (Zhou et al., 2021a) | 4Mb | real-life | electric | yes | no | transformers |
| LLA (Tay et al., 2021) | 7 700Mb | synthetic | planar paths | no | yes | transformers |
| Monash (Godahewa et al., 2021) | 501Mb | real-life | diverse | yes | no | statistical |
| Libra (Bauer et al., 2021) | 542Mb | real-life | diverse | yes | no | statistical & ML |
| Chaos (Gilpin, 2021) | 80Mb | synthetic | chaos | yes | no | statistical & NN |
| UCR (Dau et al., 2019) | 316Mb | real-life | diverse | no | yes | N/A |
| **ours** | 100 000Mb | real-life & synthetic | diverse | yes | no | NN |

## 2 DATASETS

We introduce all of the datasets included in our benchmark. Some of the datasets are scattered in various earlier works on LTSF. However, our work is the first that unifies several diverse domains within a single self-contained benchmark dataset. A summary of the synthetic datasets in our benchmark is in Tab. 2. We share unnormalized datasets in hdf5 raw format online dat. All datasets are split into training and test/evaluation sets. Data is normalized using the standard normalization with std. dev. and mean values computed over the whole training set, then test data are normalized with the pre-computed values. All synthetic datasets contain 20k trajectories, from which the first 18k are used for training and 2k are dedicated for evaluation/testing. We emphasize that the trajectories in synthetic datasets do not overlap, i.e., each is generated using a different initial condition. We provide only a training dataset and a test dataset. However, in a typical data-science workflow, it would be recommended to use the validation set as well. We only have a distinct test dataset for simplicity, but one could also split the training dataset into a training and validation set.

### 2.1 SYNTHETIC

*Sinewave.* The deterministic univariate sinewave series comprises quasi-periodic trajectories defined by $s_j = \sin(0.2 \cdot j + \phi) + \sin(0.3 \cdot j + \phi)$, where $\phi$ is a random phase $\phi \in [0, 1]$, drawn from the uniform distribution per trajectory basis. We generate trajectories by setting $j$ to a sequence of integers. We include the dataset generated by a simple deterministic periodic rule as an instance of a sanity check in our benchmark suite. The obtained results are surprisingly varied. Some methods rapidly converge to numerical zero, whereas others, including DeepAR and LSTM, fail to converge.

*Mackey-Glass.* Mackey-Glass (MG) is a family of univariate Delayed differential equation (DDE) modeling certain biological behavior (Mackey & Glass, 1977). For some range of parameters, the equation admits chaotic attractors. The equations take the form $\frac{dy}{dt} = \frac{0.2y(t-\tau)}{1+y^{10}(t-\tau)} - 0.1y(t)$, where $\tau$ is the delay which we fix to $\tau = 25$, the equation demonstrates chaotic dynamics for this particular $\tau$ value. We generate the trajectories by applying the Euler scheme with time-step 0.1 to the delayed dynamical system with uniformly random initial history per trajectory $-10\tau$ points sampled from the range $1.2 + [-0.01, 0.01]$. The main difficulty of the MG dataset lies in modeling the chaotic dynamics exhibiting high sensitivity to the initial history; the forecasting problem is especially challenging for shorter context windows.

*Lorenz.* Lorenz dynamical system is a simplified meteorological model shown to admit a chaotic attractor (the Lorenz butterfly), i.e., all trajectories are contained within a compact butterfly-shaped region but exhibit very high sensitivity to the initial condition (IC). We use the classical parameters $\sigma = 10, \rho = 28, \beta = 8/3$; the trajectories are generated by following a random IC sampled from the normal distribution centered at $(0, -0.01, 9)$ with std. dev. 0.001 using the Euler scheme with time-step 0.01. The dataset is multivariate (three variables), and the main difficulty of the Lorenz

dataset lies in modeling the chaotic dynamics exhibiting high sensitivity to the initial history; the forecasting problem is especially challenging for shorter context windows.

*Stochastic Lotka-Volterra.* The deterministic Lotka-Volterra (LV) dynamical system models the predator-prey population dynamics. It is a system of coupled nonlinear ordinary differential equations (ODEs): $\frac{dx}{dt} = \alpha x - \beta xy, \frac{dy}{dt} = \delta xy - \gamma y$, with $x, y$ denoting the prey and predator populations respectively. We set the parameters: the growth rate of prey population $\alpha = 1$, the effect of the presence of predators on the prey growth rate $\beta = 0.1$, the effect of the presence of prey on the predator's growth rate $\delta = 0.02$, the predator's death rate $\gamma = 0.5$. The initial condition is sampled uniformly per each trajectory from the range $(50, 150), (10, 30)$. Stochasticity is injected at each step by uniformly sampling $\alpha \in [1 - 0.002, 1 + 0.002]$. The deterministic LV exhibits stable periodic dynamics, but the added stochasticity perturbs the dynamics, resulting in random fluctuations, and a forecasting method aims to predict the rough trend of the population dynamics.

*Kuramoto-Sivashinsky.* We find it instructive to evaluate the models on a smooth chaotic dynamical system with a large spatial dimension (100). We chose the 1d Kuramoto-Sivashinsky (KS) fourth-order nonlinear partial differential equation (PDE) exhibiting chaotic dynamics: $u_t + u_{xx} + u_{xxxx} + \frac{1}{2}u_x^2 = 0$ with periodic boundary conditions on the interval domain $[0, 200]$. Solution $u$ is the flame velocity. Following the setup given in Linot & Graham (2022) we generated trajectories of length 1000 initiating from a random initial condition – weighted sum of $\sin(y) \& \cos(y)$, where $y \in \{x\pi/32, x\pi/16, x\pi/8, x\pi/4\}$ with uniform random weights. We used the spectral method with 100 coefficients for solving the PDE in the frequency space, applying the fast Fourier transform (FFT) and time-stepping using the Runge-Kutta (RK) scheme with $dt = 0.01$. Each trajectory is rolled out until time $t = 200$, and trajectories of length 1000 are generated, saving one in 20 frames. There are two main difficulties of the forecasting task for the KS dataset: the equation is chaotic and exhibits high sensitivity to the initial condition; the spatial dimension is large (100), requiring an efficient state encoding in a forecasting method.

*Cahn-Hillard.* Apart from chaotic dynamics, an abundance of stable physical processes is amenable to forecasting given short input sequences; such data may be of a significant spatial dimension. We choose the Cahn-Hillard (CH) PDE often encountered in research on pattern formation. The dynamics of the equation for a given parameter value and the initial condition are stable and lead to the formation of one of the stable patterns. We solve the CH equation $\partial c / \partial t = \nabla^2(c^3 - c - 0.0001 \cdot \nabla^2 c)$ in a 2D domain $\Omega = [0, 1]^2$ with periodic boundary conditions. Solution $c$ is the concentration of the fluid $c \in [-1, 1]$. The equation is solved in the frequency space using the spectral method (applying the FFT) using $64 \times 64$ grid, and the RK scheme does the time-stepping with $dt = 5e - 06$. The dataset trajectories are generated by saving each frame and recording the $16 \times 16$ uniform subgrid of each frame. The dataset consists of stable trajectories having a vast state dimension (256), requiring efficient state encoding in a forecasting method.

*MuJoCo.* We generated six datasets using the MuJoCo (Todorov et al., 2012) rigid body dynamics simulator. MuJoCo is a widely used benchmark environment in the reinforcement learning domain, where model-based approaches are based on learning a forecasting model from data for an efficient controller design. Hence, it is crucial to demonstrate guidelines for picking the suitable forecasting model in such an environment. Data is generated using three environments (Half-Cheetah, Walker2d, and Hopper). We use the D4RL framework from Fu et al. (2020). Using the pretrained policies for each task, we generate 18k training and 2k test trajectories of total length 1000. We apply the so-called medium policies from the D4RL suite that are trained up to $1/3$ of the maximal return reached by the Soft Actor-Critic (SAC) RL algorithm (an expert policy). We are ensuring that the trajectories in the dataset are diverse. Two modes of control possible using stochastic policies obtained by SAC: the deterministic actions (D) and the actions sampled randomly from the policy normal distribution (S). We obtain two datasets per environment by following the two different control modes. The MuJoCo datasets consist of trajectories exhibiting complex dynamics with a significant state dimension 11-17, obtained through an external controller.

## 2.2 REAL-LIFE

The summary of real-life datasets included in our benchmark is in Tab. 2. We emphasize that the trajectories in real-life datasets are generated using overlapping, i.e., a long trajectory is split into several overlapping sub-trajectories.

Table 2: Synthetic (top table) and real-life (bottom) datasets. All syn. datasets contain 20k trajectories in total.

| PROBLEM | DOMAIN | TRAJ. LENGTH | OBS. DIM. | LOOKBACKS | DETERM. |
|---|---|---|---|---|---|
| SINEWAVE OSCILLATOR | SIMPLE SIGNAL (SANITY CHECK) | 2000 | 1 | 2,8,96 | ✓ |
| STOCHASTIC LOTKA-VOLTERRA | SDE, POPULATION DYN. | 2000 | 2 | 96, 500, 2000 | ✗ |
| MACKEY-GLASS | DDE, CHAOS | 2000 | 1 | 96,500,1000 | ✓ |
| LORENZ | ODE, CHAOS | 2000 | 3 | 96,500,1000 | ✓ |
| MUJOCO HOPPER D | RL, ROBOTIC SIM. | 1000 | 11 | 96, 250, 500 | ✓ |
| MUJOCO CHEETAH D | RL, ROBOTIC SIM. | 1000 | 17 | 96, 250, 500 | ✓ |
| MUJOCO WALKER2D D | RL, ROBOTIC SIM. | 1000 | 17 | 96, 250, 500 | ✓ |
| MUJOCO HOPPER S | RL, ROBOTIC SIM. | 1000 | 11 | 96, 250, 500 | ✗ |
| MUJOCO CHEETAH S | RL, ROBOTIC SIM. | 1000 | 17 | 96, 250, 500 | ✗ |
| MUJOCO WALKER2D S | RL, ROBOTIC SIM. | 1000 | 17 | 96, 250, 500 | ✗ |
| KURAMOTO-SIVASHINSKY PDE | PDE, CHAOS | 1000 | 100 | 96, 250, 500 | ✓ |
| CAHN-HILLARD PDE | PDE, STABLE PATTERN FORMATION | 1000 | 256 | 96, 250, 500 | ✓ |

| PROBLEM | DOMAIN | TOTAL # TRAJS. | TRAJ. LENGTH | OBS. DIM. | LOOKBACKS |
|---|---|---|---|---|---|
| M4 | FINANCE | 289094 | 247 | 1 | 96, 168 |
| ETTM $1+2$ | ELECTRICITY | 133602 | 1440 | 1 | 96, 336, 720 |
| ELECTRICITY | ELECTRICITY | 1083452 | 1440 | 1 | 96, 336, 720 |
| ETTM $1+2$ (LONG) | ELECTRICITY | 131362 | 2000 | 1 | 1000 |
| WEATHER | WEATHER | 208295 | 1000 | 21 | 96, 250, 500 |
| PEMS-BAY | TRAFFIC | 51542 | 288 | 325 | 144 |

*M4.* We took weekly time series from the M4 competition dataset (Makridakis et al., 2020), a financial data with exact origin not revealed. We preprocessed the raw data. We filtered out these series shorter than 247 (It gave us a good balance between the number of series and their length). We have used the first 240 trajectories as the training dataset and the rest as the test dataset (variable length). For every trajectory in the original dataset, we included all 247-length overlapping sub-trajectories in the respective final dataset (train/test). We do not include the newer M5 dataset, as we found that M5 would be more suited towards a hand-crafted forecasting procedure (inc., hierarchical forecasting and specific feature engineering) than to our current effort.

*ETTm $1+2$.* We have used two series from the ETT-small dataset (m1 and m2) introduced by Zhou et al. (2021b). The data is recorded from electrical transformers gathered in two regions of China in 15-minute intervals. First, We have divided them into the training and test datasets (initial 80% going into the training dataset). The series was then divided into lengths 1440 in the same fashion as in the M4 dataset. Moreover, we use a long variant with trajectories of length 2000. There are seven variables in the dataset. The most popular option is forecasting one target variable (the remaining six are exogenous).

*Electricity.* We have used the 370 series from the Household Electric Power Consumption data (ele). They were divided the same way as the M4 dataset (trajectories of length 1440, with training/test datasets split on at 2014-08-31 23:00). We used a random $1/8$ of the resulting trajectories for training and testing.

*Weather.* We have used the daily records of weather data (21 measurements) shared in wea. We merged the data from years 2019 to 2022 inclusive into one trajectory and divided it into overlapping subtrajectories of length 1000.

*PEMS-BAY.* Traffic dataset collected by California Transportation Agencies (CalTrans) Performance Measurement System (PEMS) used in Li et al. (2018). Selected 325 sensors in the Bay Area (BAY), collected during 6 first months of 2017. The dataset has the largest state dimension of all of the included datasets.

## 2.3 BENCHMARK SYNERGY

Our dataset comprises two main components: real-life datasets commonly used in the recent literature, the PEMS-BAY traffic dataset, contrary to the standard datasets, characterized by a large number of interacting channels, and a collection of clean, diverse synthetic datasets on top of that. These datasets serve as a reference point and provide a foundation for benchmarking the performance of LTSF ML models. Complementing the real-life datasets, we have included a set of synthetic datasets that pose different challenges. Tab. 2 shows that our synthetic collection complements the real-life datasets in several aspects (lengths, obs. dim., degree of stochasticity). We leverage the flexibility of a broad spectrum of synthetic datasets to exhibit significantly longer trajectories,

with up to 2k steps, and multi-variate in nature. Furthermore, the synthetic datasets encompass simulated dynamical systems, such as the rigid body dynamics of robots using MuJoCo, which is particularly relevant for control design. We incorporate two modes of control, deterministic and stochastic, to account for different control strategies. Moreover, the systems belong into two distinct modes: stable dynamics (e.g., L.-V., C.-H. PDE) and chaotic dynamics (e.g., M.-G., Lorenz, K.-S. PDE). By combining real-life and synthetic datasets, we believe our benchmark dataset provides a comprehensive evaluation framework to assess the performance of NN and, in general, ML models in time-series forecasting. Including diverse scenarios enables a more robust analysis and facilitates the exploration of model generalization across different applications. The prescribed lookback lengths are adjusted per dataset basis and were chosen based on the setting studied in previous work, i.e., short, medium, and long lookback lengths are set. Refer to Sec. B for descriptive statistics.

## 3 BENCHMARKED MACHINE LEARNING MODELS

We briefly present the state-of-the-art NN-based models that we benchmarked. We do not include spatio-temporal graph models, which may work well with datasets such as PEMS-BAY, but this was the only dataset we used with a clear graph representation. All of the custom implementation improvements are provided. We share a self-contained PyTorch library comprising the standardized implementations of all the models (except the PatchTST, for which we used the original repository) in the supplement. Tables with the used hyperparameters can be found in the appendix. In most cases, we relied on the original hyperparameters provided by the authors, in some cases requiring slight tuning. PDE problems and the longest forecasting horizons are infeasible for some models (excluded from Tab. 3).

*LSTM (Hochreiter & Schmidhuber, 1997).* The long-short term memory (LSTM) model is a special kind of recurrent NN capable of learning long-term dependencies in data. We train the model, using one LSTM to encode the lookback to a latent state and the other to predict new states. We include LSTM as a baseline, demonstrating strong performance for some datasets.

*DeepAR (Salinas et al., 2020).* DeepAR is an LSTM-based recurrent NN trained in the autoregressive fashion contrary to the vanilla LSTM model. The original algorithm is trained to output a probability distribution, which may be used as, e.g., uncertainty estimates. However, we used a modified DeepAR implementation outputting point estimates. Apart from that, we train the model backpropagating through the whole horizon, automatically using a prediction from timestep $t$ as input for timestep $t + 1$.

*DeepAR + CL (ours).* We implement our customized DeepAR model enhanced with a curriculum learning phase. We adapt the widely known curriculum learning technique (Graves et al., 2017; Cirik et al., 2016) for TS forecasting. Before the actual model training on the full-length trajectories, a 'warm-start' phase is first performed in which the encoder and the model are pretrained on shorter trajectories of gradually increasing lengths. We demonstrate that such modified training consistently improves the performance of the vanilla DeepAR.

*Latent ODE (Rubanova et al., 2019).* Neural ODEs (Chen et al., 2018) is one of the most influential families of continuous recurrent NN models. Neural ODEs define a time-dependent hidden state $h(t)$ as the solution to the initial value problem $\frac{dh(t)}{dt} = f_\theta(h(t), t), h(t_0) = t_0$ and $f_\theta$ is a NN parametrized by a vector $\theta$. We used a nongenerative setting, a smaller architecture than the original work, with an LSTM encoder and an multi-layer perceptron (MLP) decoder. Nonetheless, the LTSF setting is still challenging for NODE models due to the sequential nature of ODE solvers.For real-life datasets, we have used the same curriculum learning technique as for DeepAR, which also significantly improved its performance. We are training the model in a standard non-VAE fashion.

*N-Hits (Challu et al., 2022).* An improvement of the earlier model N-Beats by Oreshkin et al. (2020). Incorporating hierarchical interpolation and multi-rate data sampling techniques, N-HiTS sequentially assembles predictions while emphasizing different frequency components and scales. This approach effectively decomposes the input signal and synthesizes forecasts. We have modified the original implementation to add support for multivariate time series.

*LTSF NLinear (Zeng et al., 2023).* As described by the authors, an 'embarrassingly simple one-layer linear model' was demonstrated to outperform the sophisticated transformer-based models in real-life datasets. LTSF Linear models are trained in sequence-to-sequence fashion and operate directly on

the states; hence, they are challenging to apply for datasets with larger dimensional states. One modification that we have applied is working on the flattened states, i.e. model is not channel independent and thus is very parameter heavy for high-dimensional data.

*Latent LTSF NLinear (ours).* In order to circumvent the limitations of LTSF Linear models about restricted state-space dimension and modeling of nonlinear dependencies between the state components, we introduce the latent LTSF NLinear model. The linear map is applied to the latent representations of states instead of the states directly. Our model outperforms the vanilla LTSF NLinear in most benchmarked datasets and can be applied to problems with large spatial dimensions (PDEs). We used an LSTM encoder and an MLP decoder. To limit the parameter count, the latent dimension is set to be smaller than the observation space for a high-dimensional series.

*SpaceTime (Zhang et al., 2023).* A recent State-space model (SSM) is dedicated to effective time-series modeling. SSMs are classical models for time series, and prior works combine SSMs with deep learning layers for efficient sequence modeling (S4 and subsequent models). It can express complex dependencies, forecast over long horizons, and efficiently train over long sequences. Contrary to the S4 model, it can model autoregressive processes.

*Patch Transformer (Nie et al., 2023) (PatchTST).* At the time of submission, this is the newest model from the LTSF transformer family, shown to outperform the earlier transformer models for time-series forecasting and the LTSF-Linear(Zeng et al., 2023). It is based on segmenting time series into subseries-level patches, which serve as input tokens. We have modified the original implementation to allow for interactions between series variables.

## 4 BENCHMARK RESULTS

We present the thorough NN model benchmark results on the introduced dataset. Space considerations defer some of our experimental results to the appendix. In all subsequent tables, minimal mean-squared error (MSE) and mean absolute error (MAE) metrics were reported achieved during training on the test set (2k trajectories). The best results are marked with boldface and blue color. We report results rounded to two decimal places. We share spreadsheets with precise results in the supplement.

We now describe how the LTSF task is set up. Given $L < N$ *lookback window* size (also called the history). Let $X = (s_0, s_1, \ldots, s_{L-1})$, $Y = (s_L, \ldots, s_N)$ be a trajectory not seen during the training phase split into the initial chunk of length $L$ (the lookback window length) and the remainder of length $T = N - L$ (the future). The task of a forecasting model is to predict $\hat{Y}$ given $X$ such that $\|\hat{Y} - Y\|$ is minimized, where $\| \cdot \|$ is either the usual MSE or MAE metric. We set the lookback windows based on the setting studied in previous work, i.e., short, medium, and long lookback lengths are used, where the longest lookback length is set to the first half of the total trajectory length, and the shortest is less than 10% of the total trajectory length. Benchmark results are in Tab. 3.and Tab. 4.

### 4.1 COMPUTATIONAL METHODOLOGY

Each of the experiments was performed on an RTX2080Ti GPU card or equivalent. Each experiment was run until convergence by visual investigation of the loss curves or until reaching the global 8 hours cap. We utilized a modest computational cluster in an academic setting comprised of 16 GPUs. The whole benchmark required performing 335 runs, giving 2680 GPU hours in total. The rate of performing optimization epochs and hence the convergence rate is highly model specific. To account for varied convergence speeds, we set a global runtime cap uniform for all datasets equal to 8 hours. If the metric values are not reported for the given model in Sec. 4, it means that it was infeasible to perform such an evaluation.

## 5 BENCHMARK CONCLUSIONS

We present our paper's key findings, which we believe hold significant implications for the machine learning community and offer valuable insights for future developments in research on NN models dedicated to LTSF. See Fig. 1 for bar plots visualizing the benchmark results averaged over classes.

*Need of sanity check datasets.* As demonstrated by our sanity-check dataset – the Sinewave (results in Tab. 3), surprisingly, such a simple dataset diversifies the NN models. Not all of the models performing well in more complicated scenarios managed to converge here on much simpler signal data. In particular, LSTM and DeepAR models stand out and struggle to converge even in the case of

Table 3: Benchmark Results on the synthetic datasets described in Sec. 2.1. The resulting metrics are shown rounded to two decimal places. The numbers shown in the first table were also multiplied by 100.

| DATASET | L | N-HITS MSE | MAE | SPACETIME MSE | MAE | LATENT NLINEAR MSE | MAE | NLINEAR MSE | MAE | DEEPAR CL MSE | MAE | LSTM MSE | MAE | DEEPAR VANILLA MSE | MAE | NEURAL ODE MSE | MAE | PATCHT MSE | MAE |
|---|---|---|---|---|---|---|---|---|---|---|---|---|---|---|---|---|---|---|---|
| SIN. | 1 | *0.0* | *0.0* | 0.0 | 0.8 | 0.1 | 1.8 | 33.2 | 44.1 | 69.0 | 70.1 | 87.5 | 71.2 | 98.3 | 79.9 | 98.8 | 80.3 | 607.1 | 188.2 |
| | 2 | *0.0* | *0.0* | 0.0 | 0.6 | 0.0 | 0.2 | 1.8 | 10.2 | 53.6 | 63.4 | 79.9 | 66.7 | 98.9 | 80.2 | 98.9 | 80.3 | 1071.7 | 253.1 |
| | 8 | 0.0 | 0.0 | 0.0 | 0.4 | 0.0 | 0.0 | *0.0* | *0.0* | 3.0 | 13.3 | 85.2 | 70.1 | 99.0 | 80.4 | 99.7 | 81.0 | 0.0 | 1.2 |
| | 96 | 0.0 | 0.0 | 0.0 | 0.9 | 0.0 | 0.0 | *0.0* | *0.0* | 3.7 | 14.9 | 77.0 | 63.1 | 99.6 | 81.0 | 100.5 | 81.4 | 0.1 | 1.5 |
| AVG. | | *0.0* | *0.0* | 0.0 | 0.7 | 0.0 | 0.5 | 8.7 | 13.6 | 32.3 | 40.4 | 82.4 | 67.8 | 98.9 | 80.4 | 99.5 | 80.7 | 419.7 | 111.0 |

| DATASET | L | LSTM MSE | MAE | N-HITS MSE | MAE | LATENT NLINEAR MSE | MAE | DEEPAR CL MSE | MAE | SPACETIME MSE | MAE | NEURAL ODE MSE | MAE | NLINEAR MSE | MAE | DEEPAR VANILLA MSE | MAE | PATCHT MSE | MAE |
|---|---|---|---|---|---|---|---|---|---|---|---|---|---|---|---|---|---|---|---|
| L.-V. | 96 | *0.80* | *0.61* | 0.83 | 0.63 | 0.81 | 0.61 | 0.81 | 0.62 | 0.83 | 0.63 | 0.90 | 0.68 | 0.89 | 0.68 | 0.89 | 0.67 | 0.99 | 0.69 |
| | 500 | *0.78* | 0.59 | 0.80 | 0.61 | 0.79 | 0.59 | 0.79 | *0.59* | 0.81 | 0.63 | 0.87 | 0.66 | 0.84 | 0.64 | 0.87 | 0.65 | 0.95 | 0.66 |
| | 1000 | *0.63* | *0.49* | 0.71 | 0.55 | 0.70 | 0.53 | 0.64 | 0.50 | 0.87 | 0.67 | 0.78 | 0.60 | 0.87 | 0.67 | 0.76 | 0.58 | 0.82 | 0.58 |
| M.-G. | 96 | 0.67 | 0.59 | *0.64* | *0.55* | 0.68 | 0.58 | 0.80 | 0.69 | 0.74 | 0.64 | 0.96 | 0.79 | 0.82 | 0.70 | 1.00 | 0.82 | 0.77 | 0.64 |
| | 500 | *0.66* | *0.58* | 0.74 | 0.63 | 0.80 | 0.67 | 0.70 | 0.62 | 0.81 | 0.71 | 0.88 | 0.76 | 0.90 | 0.76 | 0.98 | 0.81 | 0.88 | 0.73 |
| | 1000 | *0.49* | *0.46* | 0.73 | 0.64 | 0.78 | 0.66 | 0.96 | 0.60 | 0.99 | 0.82 | 0.86 | 0.75 | 0.92 | 0.77 | 0.96 | 0.80 | 0.86 | 0.72 |
| LORENZ | 96 | 0.56 | 0.51 | *0.48* | *0.43* | 0.54 | 0.49 | 0.61 | 0.55 | 0.63 | 0.57 | 0.76 | 0.67 | 0.69 | 0.60 | 0.66 | 0.59 | 0.70 | 0.55 |
| | 500 | 0.60 | 0.54 | *0.58* | *0.52* | 0.61 | 0.53 | 0.67 | 0.60 | 0.76 | 0.68 | 0.84 | 0.74 | 0.84 | 0.73 | 0.90 | 0.78 | 0.94 | 0.74 |
| | 1000 | *0.47* | *0.43* | 0.67 | 0.59 | 0.71 | 0.62 | 0.83 | 0.63 | 0.97 | 0.82 | 0.80 | 0.70 | 0.88 | 0.75 | 0.65 | 0.60 | 1.05 | 0.81 |
| AVG. | | *0.63* | *0.53* | 0.69 | 0.57 | 0.71 | 0.59 | 0.76 | 0.60 | 0.82 | 0.69 | 0.85 | 0.71 | 0.85 | 0.70 | 0.85 | 0.70 | 0.88 | 0.68 |

| DATASET | L | DEEPAR CL MSE | MAE | LSTM MSE | MAE | SPACETIME MSE | MAE | LATENT NLINEAR MSE | MAE | N-HITS MSE | MAE | NLINEAR MSE | MAE | DEEPAR VANILLA MSE | MAE | NEURAL ODE MSE | MAE | PATCHT MSE | MAE |
|---|---|---|---|---|---|---|---|---|---|---|---|---|---|---|---|---|---|---|---|
| CHEETAH(S) | 96 | *0.79* | *0.71* | 0.80 | 0.72 | 0.80 | 0.72 | 0.80 | 0.72 | 0.82 | 0.73 | 0.81 | 0.73 | 0.90 | 0.79 | 0.91 | 0.80 | 0.91 | 0.77 |
| | 250 | *0.76* | *0.69* | 0.77 | 0.70 | 0.78 | 0.71 | 0.78 | 0.71 | 0.82 | 0.73 | 0.80 | 0.72 | 0.95 | 0.82 | 0.95 | 0.82 | 0.89 | 0.75 |
| | 500 | *0.68* | *0.64* | 0.68 | 0.65 | 0.70 | 0.66 | 0.70 | 0.66 | 0.77 | 0.69 | 0.73 | 0.68 | 0.94 | 0.82 | 0.89 | 0.79 | 0.80 | 0.70 |
| HOPPER(S) | 96 | 0.72 | *0.48* | *0.72* | 0.48 | 0.73 | 0.49 | 0.73 | 0.48 | 0.74 | 0.49 | 0.75 | 0.51 | 0.72 | 0.48 | 0.74 | 0.50 | 1.14 | 0.63 |
| | 250 | 0.75 | 0.48 | *0.74* | *0.48* | 0.77 | 0.50 | 0.77 | 0.50 | 0.79 | 0.52 | 0.81 | 0.53 | 0.75 | 0.48 | 0.79 | 0.52 | 1.16 | 0.64 |
| | 500 | *0.63* | 0.44 | 0.63 | *0.44* | 0.67 | 0.47 | 0.68 | 0.48 | 0.69 | 0.50 | 0.73 | 0.52 | 0.65 | 0.45 | 0.68 | 0.48 | 0.95 | 0.58 |
| WALKER(S) | 96 | 0.86 | 0.65 | *0.86* | *0.64* | 0.87 | 0.65 | 0.87 | 0.65 | 0.88 | 0.65 | 0.88 | 0.66 | 0.87 | 0.65 | 0.87 | 0.66 | 1.28 | 0.79 |
| | 250 | *0.85* | 0.62 | 0.85 | *0.62* | 0.87 | 0.64 | 0.89 | 0.65 | 0.91 | 0.67 | 0.94 | 0.70 | 0.85 | 0.62 | 0.90 | 0.66 | 1.30 | 0.80 |
| | 500 | *0.68* | 0.50 | 0.69 | 0.50 | 0.76 | 0.57 | 0.75 | 0.56 | 0.80 | 0.59 | 0.83 | 0.63 | 0.69 | *0.50* | 0.73 | 0.54 | 1.06 | 0.69 |
| AVG. | | *0.75* | *0.58* | 0.75 | 0.58 | 0.77 | 0.60 | 0.77 | 0.60 | 0.80 | 0.62 | 0.81 | 0.63 | 0.81 | 0.62 | 0.83 | 0.64 | 1.06 | 0.71 |

| DATASET | L | PATCHT MSE | MAE | SPACETIME MSE | MAE | DEEPAR CL MSE | MAE | LSTM MSE | MAE | LATENT NLINEAR MSE | MAE | DEEPAR VANILLA MSE | MAE |
|---|---|---|---|---|---|---|---|---|---|---|---|---|---|
| K.-S. | 96 | 1.05 | 0.85 | 0.97 | 0.81 | *0.92* | *0.78* | 0.96 | 0.81 | 0.99 | 0.82 | 0.96 | 0.81 |
| | 250 | 1.06 | 0.85 | 0.97 | 0.82 | *0.90* | *0.77* | 0.97 | 0.81 | 1.00 | 0.83 | 0.97 | 0.81 |
| | 500 | 1.04 | 0.84 | 0.97 | 0.82 | *0.86* | *0.74* | 0.94 | 0.79 | 0.99 | 0.82 | 0.93 | 0.79 |
| C.-H. | 96 | *0.46* | *0.52* | 0.57 | 0.63 | 1.01 | 0.89 | 0.74 | 0.71 | 0.83 | 0.78 | 1.00 | 0.88 |
| | 250 | *0.36* | *0.45* | 0.49 | 0.58 | 0.59 | 0.64 | 0.73 | 0.71 | 0.87 | 0.79 | 1.00 | 0.86 |
| | 500 | *0.27* | *0.39* | 0.79 | 0.74 | 0.50 | 0.57 | 0.67 | 0.66 | 0.89 | 0.80 | 1.17 | 0.97 |
| AVG. | | *0.71* | *0.65* | 0.79 | 0.73 | 0.80 | 0.73 | 0.84 | 0.75 | 0.93 | 0.81 | 1.01 | 0.85 |

the larger lookback (96). We note that CL alleviates the issue to some extent. PatchTST diverges for short lookbacks (1, 2) but still converges for larger.

*Best models depend on the dataset.* The notion that newer models outperform older ones is challenged by our findings. While the newer models (N-Hits, NLinear, SpaceTime, PatchTST) dominate the older LSTM and DeepAR approaches in univariate real-life data (see Tab.4), the situation changes dramatically with multivariate and synthetic datasets. When focusing on low-dimensional, long-trajectory datasets (as shown in Tab.3), LSTM performs the best, with N-Hits proving competitive for chaotic datasets, see also (Gilpin, 2021). In the case of the previously unexplored MuJoCo data, classical LSTM and DeepAR (with CL) models exhibit the best performance.

*Underappreciated baselines: Classical NN models.* We emphasize that the classical approaches, LSTM and DeepAR, have often been overlooked as baselines. However, our experiments reveal their consistently strong performance compared to state-of-the-art models. Therefore, we argue that

Table 4: Benchmark Results on the real-life datasets described in Sec. 2.2. The resulting metrics are shown rounded to two decimal places.

| DATASET | L | N-HITS MSE | N-HITS MAE | LATENT NLINEAR MSE | LATENT NLINEAR MAE | NLINEAR MSE | NLINEAR MAE | PATCHT MSE | PATCHT MAE | SPACETIME MSE | SPACETIME MAE | LSTM MSE | LSTM MAE | DEEPAR CL MSE | DEEPAR CL MAE | DEEPAR VANILLA MSE | DEEPAR VANILLA MAE | NEURAL ODE MSE | NEURAL ODE MAE |
|---|---|---|---|---|---|---|---|---|---|---|---|---|---|---|---|---|---|---|---|
| ETTM1+2 | 96 | 0.21 | 0.33 | 0.22 | 0.33 | 0.23 | 0.35 | 0.22 | 0.34 | *0.21* | *0.33* | 0.24 | 0.35 | 0.23 | 0.34 | 0.35 | 0.44 | 0.32 | 0.41 |
|  | 336 | 0.17 | *0.30* | 0.18 | 0.30 | 0.19 | 0.32 | *0.17* | 0.31 | 0.17 | 0.31 | 0.21 | 0.33 | 0.18 | 0.32 | 0.22 | 0.34 | 0.33 | 0.44 |
|  | 720 | 0.15 | 0.29 | *0.15* | *0.28* | 0.16 | 0.29 | 0.15 | 0.29 | 0.17 | 0.31 | 0.17 | 0.30 | 0.18 | 0.31 | 0.19 | 0.32 | 0.96 | 0.78 |
| M4 | 96 | 0.21 | 0.22 | 0.22 | 0.22 | 0.25 | 0.25 | 0.21 | *0.21* | *0.20* | 0.22 | 0.22 | 0.23 | 0.23 | 0.24 | 0.23 | 0.24 | 0.24 | 0.25 |
|  | 168 | *0.12* | 0.16 | 0.14 | 0.16 | 0.14 | 0.17 | 0.13 | *0.15* | 0.12 | 0.16 | 0.14 | 0.16 | 0.13 | 0.17 | 0.13 | 0.16 | 0.15 | 0.19 |
| ELECTRIC | 96 | 0.30 | 0.05 | 0.29 | 0.05 | 0.35 | 0.05 | 0.36 | 0.06 | *0.15* | *0.05* | 0.91 | 0.10 | 0.99 | 0.10 | 1.52 | 0.17 | 4.55 | 1.58 |
|  | 336 | 0.21 | 0.05 | 0.19 | *0.04* | 0.26 | 0.05 | 0.32 | 0.05 | *0.13* | 0.05 | 0.72 | 0.10 | 0.85 | 0.10 | 1.26 | 0.21 | 4.73 | 1.28 |
|  | 720 | 0.17 | 0.04 | 0.19 | 0.04 | *0.11* | *0.03* | 0.19 | 0.04 | 0.25 | 0.07 | 0.70 | 0.09 | 0.71 | 0.10 | 1.13 | 0.18 | 0.82 | 0.12 |
| ETTM (LONG) | 1000 | 0.17 | 0.30 | *0.16* | *0.29* | 0.16 | 0.30 | 0.16 | 0.30 | 1.02 | 0.92 | 0.19 | 0.32 | 0.19 | 0.32 | 0.32 | 0.43 | 0.30 | 0.41 |
| AVG. |  | *0.19* | 0.19 | 0.19 | *0.19* | 0.21 | 0.20 | 0.21 | 0.20 | 0.27 | 0.27 | 0.39 | 0.22 | 0.41 | 0.22 | 0.59 | 0.28 | 1.38 | 0.61 |

| DATASET | L | LATENT NLINEAR MSE | LATENT NLINEAR MAE | LSTM MSE | LSTM MAE | SPACETIME MSE | SPACETIME MAE | DEEPAR CL MSE | DEEPAR CL MAE | DEEPAR VANILLA MSE | DEEPAR VANILLA MAE | PATCHT MSE | PATCHT MAE | N-HITS MSE | N-HITS MAE | NLINEAR MSE | NLINEAR MAE | NEURAL ODE MSE | NEURAL ODE MAE |
|---|---|---|---|---|---|---|---|---|---|---|---|---|---|---|---|---|---|---|---|
| PEMS-BAY | 144 | 0.68 | 0.41 | *0.67* | *0.36* | 0.69 | 0.39 | 0.71 | 0.38 | 0.73 | 0.38 | N/A | N/A | N/A | N/A | N/A | N/A | 0.76 | 0.41 |
| WEATHER | 96 | *0.71* | *0.43* | 0.72 | 0.43 | 0.73 | 0.45 | 0.75 | 0.45 | 0.88 | 0.54 | 0.91 | 0.46 | 0.82 | 0.47 | 0.98 | 0.49 | 1.66 | 0.84 |
|  | 250 | *0.69* | 0.42 | 0.69 | *0.42* | 0.71 | 0.43 | 0.75 | 0.45 | 0.87 | 0.54 | 0.81 | 0.43 | 0.85 | 0.46 | 0.86 | 0.47 | 0.85 | 0.53 |
|  | 500 | *0.66* | 0.41 | 0.67 | 0.41 | 0.69 | 0.43 | 0.72 | 0.43 | 0.73 | 0.44 | 0.72 | *0.40* | 0.83 | 0.45 | 0.76 | 0.45 | 0.66 | 0.41 |
| AVG. |  | *0.68* | 0.42 | 0.69 | *0.41* | 0.71 | 0.42 | 0.73 | 0.43 | 0.80 | 0.47 | 0.81 | 0.43 | 0.83 | 0.46 | 0.86 | 0.47 | 0.98 | 0.55 |

(a) MSE averaged over chaotic and MuJoCo datasets

(b) MSE averaged over univariate real-life datasets

(c) MSE averaged over the Weather dataset results

Figure 1: Barplots illustrating the relative performance of the benchmarked methods over synthetic and real-life datasets (taking into account those feasible by the entirety of methods only). Note Fig. 1a and 1c are rescaled.

these models deserve to be included in the evaluated baselines, particularly when addressing LTSF scenarios beyond the standard set of univariate real-life datasets.

*DeepAR + CL and Latent LTSF models are competitive.* We emphasize that DeepAR + CL and Latent LTSF models beat their vanilla counterparts in almost the entire benchmark. CL opens the possibility of applying DeepAR in the LTSF setting, and the Latent NLinear variant opens the possibility of applying Linear models in problems with high space dimensions. To the best of our knowledge, our work is the first to evaluate and demonstrate the high performance of these models across various LTSF scenarios, offering strong competition to the recently introduced state-of-the-art approaches.

## 6 FUTURE WORK AND CONCLUSIONS

We proposed a new LTSF benchmark dataset and extensive benchmark of the current state-of-the-art. We are convinced that our contribution holds implications for the machine learning community and offers valuable insights for future developments in research on the ML approach to LTSF. Each benchmarked model involves its own set of hyperparameters, and it was out of this work's scope to fine-tune them carefully. Performed benchmark provides first insights on the performance of the models for default or close to default hyper-parameters and the method robustness to hyperparameter values. Extensive fine-tuning the models for better performance in diverse scenarios is an important future research direction.

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

## A    APPENDIX

### A.1    DATASET HOSTING, LICENSING, AND MAINTENANCE

We share our dataset under the CC license. All of the synthetic datasets were generated from scratch by us using tailored scripts that will be shared upon publication or using available open-source repositories. We include and reformat some real-life datasets that are also shared under CC license, we appropriately cite the original work. The dataset can be accessed on the gdrive folder for the purpose of submission. Upon publication, the paper will be moved to persistent departmental storage.

### A.2    DATASET CONVENTIONS

Datasets are shared as hdf5 files dat, which we find a convenient format to store several tensors within a single file. Ech of the dataset hdf's has two keys: `train_data` and `test_data` for accessing the training trajectories and test trajectories respectively. We follow the convention of storing multiple trajectories and dimensions within a single three dimensional tensor. The first dimension corresponds to a trajectory, the second dimension to a time-stamp, and the third dimension to the state coordinate. Example `train_data` shape for the cheetah dataset reads `(18000, 1000, 17)`, i.e. there are 18k trajectories of length 1000, states are 17 dimensional. Data in the hdf files are always unnormalized, we perform data normalization during the loading process using statistics of the training data portion. We have followed established methods of generating the synthetic datasets, such as integration with common numerical methods or standard frameworks for Reinforcement Learning. Additionally, we have visually inspected a random set of trajectories from the dataset to search for potential errors. Missing values are absent entirely, as our generating code detects `NaNs` in case they appear during the generation procedure, which would indicate numerical issues. Statistical descriptors of our datasets per variable basis are provided in Sec. B.

### A.3    ADDITIONAL EXPERIMENTS

We present some additional experiments that were deferred from the main part of the paper.

#### A.3.1    PERFORMANCE IN A SCARCE TRAINING TRAJECTORY REGIME

Our benchmark datasets consist of a relatively large number of training trajectories (all synthetic datasets consists of 20k trajectories). Evaluating the NN models in a regime of scarce training trajectories is also worth investigating. We performed an experiment in which the training dataset was reduced to only 1000 trajectories and evaluated the NN models in such a scenario for a selection of datasets. We observe that in such a setting, metrics change significantly compared to the results obtained for the full datasets reported in Sec. 4. Generally, in the case of synthetic datasets presented in Tab. 5, the SpaceTime model is now stronger. Latent NLinear still outperforms its vanilla counterpart, whereas DeepAR with curriculum learning is now less performant than vanilla DeepAR. Whereas in case of the real-life datasets vanilla NLinear and N-Hits performs better, and similarly DeepAR with curriculum learning is now less performant than vanilla DeepAR. It is imporant to note, that NLinear and Latent NLinear here use a different algorithm, that performs a linear transformation of the lookback and adds the results to the last observation in the lookback (instead of performing the transformation on the lookback sequence that had the value last observation subtracted from each state).

#### A.3.2    MUJOCO D BENCHMARK

Results are presented in Tab. 7. Quantitatively the results are similar to those for MuJoCo S datasets, hence were omitted from the main part of the paper.

#### A.3.3    XGBOOST

We have conducted experiments that use the XGBoost (Chen & Guestrin, 2016) algorithm instead of a NN-based model on a few datasets. The direct approach was used, which inputs the whole lookback as separate variables (flattened multivariate observations) to the model, which outputs the whole horizon at once. Other approaches yielded similar or worse test results. It is important to

Table 5: Benchmark Results on the synthetic datasets in the scarce training trajectories regime, *we limited the training sets to only* 1000 *trajectories and have used shorter time limits*. The resulting metrics are shown rounded to two decimal places. The numbers shown in the first table were also multiplied by 100.

| DATASET | L | NLINEAR MSE | MAE | N-HITS MSE | MAE | LATENT NLINEAR MSE | MAE | SPACETIME MSE | MAE | PATCHTST MSE | MAE | DEEPAR CL MSE | MAE | LSTM MSE | MAE | DEEPAR VANILLA MSE | MAE | NEURAL ODE MSE | MAE |
|---|---|---|---|---|---|---|---|---|---|---|---|---|---|---|---|---|---|---|---|
| SIN. | 8 | *0.0* | *0.0* | 0.0 | 0.0 | 0.0 | 0.0 | 0.0 | 0.4 | 2.0 | 11.1 | 58.0 | 66.1 | 74.2 | 67.7 | 99.2 | 80.6 | 99.5 | 80.8 |
|  | 96 | *0.0* | *0.0* | 0.0 | 0.0 | 0.0 | 0.0 | 0.0 | 0.7 | 0.2 | 3.7 | 6.1 | 18.2 | 88.8 | 74.4 | 96.7 | 79.5 | 100.3 | 81.3 |
| AVG. |  | *0.0* | *0.0* | 0.0 | 0.0 | 0.0 | 0.0 | 0.0 | 0.6 | 1.1 | 7.4 | 32.0 | 42.1 | 81.5 | 71.1 | 97.9 | 80.0 | 99.9 | 81.1 |

| DATASET | L | N-HITS MSE | MAE | LATENT NLINEAR MSE | MAE | SPACETIME MSE | MAE | NLINEAR MSE | MAE | DEEPAR VANILLA MSE | MAE | NEURAL ODE MSE | MAE | LSTM MSE | MAE | PATCHTST MSE | MAE | DEEPAR CL MSE | MAE |
|---|---|---|---|---|---|---|---|---|---|---|---|---|---|---|---|---|---|---|---|
| M.-G. | 96 | *0.74* | *0.62* | 0.77 | 0.64 | 0.76 | 0.66 | 0.83 | 0.71 | 0.99 | 0.81 | 0.98 | 0.81 | 0.99 | 0.82 | 0.84 | 0.69 | 0.93 | 0.78 |
|  | 500 | 0.88 | 0.73 | 0.92 | 0.76 | *0.83* | *0.72* | 0.94 | 0.78 | 0.97 | 0.81 | 0.96 | 0.80 | 0.96 | 0.80 | 1.01 | 0.81 | 0.99 | 0.81 |
|  | 1000 | *0.91* | *0.76* | 0.96 | 0.79 | 0.99 | 0.82 | 1.02 | 0.81 | 0.96 | 0.80 | 0.97 | 0.80 | 0.96 | 0.80 | 1.06 | 0.84 | 0.97 | 0.80 |
| L.-V. | 96 | 0.88 | *0.66* | 0.89 | 0.66 | *0.87* | 0.67 | 0.90 | 0.69 | 1.00 | 0.75 | 0.99 | 0.71 | 0.95 | 0.73 | 1.08 | 0.74 | 1.00 | 0.74 |
|  | 500 | 0.84 | *0.63* | 0.87 | 0.64 | *0.83* | 0.65 | 0.88 | 0.66 | 0.86 | 0.65 | 0.98 | 0.73 | 0.95 | 0.72 | 1.06 | 0.72 | 1.53 | 0.84 |
|  | 1000 | *0.80* | *0.60* | 0.91 | 0.66 | 1.05 | 0.77 | 0.95 | 0.70 | 0.98 | 0.70 | 0.94 | 0.68 | 1.00 | 0.73 | 1.16 | 0.77 | 1.43 | 0.79 |
| AVG. |  | *0.84* | *0.67* | 0.89 | 0.69 | 0.89 | 0.72 | 0.92 | 0.72 | 0.96 | 0.75 | 0.97 | 0.76 | 0.97 | 0.77 | 1.03 | 0.76 | 1.14 | 0.79 |

| DATASET | L | LSTM MSE | MAE | SPACETIME MSE | MAE | LATENT NLINEAR MSE | MAE | DEEPAR VANILLA MSE | MAE | DEEPAR CL MSE | MAE | N-HITS MSE | MAE | NEURAL ODE MSE | MAE | NLINEAR MSE | MAE | PATCHTST MSE | MAE |
|---|---|---|---|---|---|---|---|---|---|---|---|---|---|---|---|---|---|---|---|
| CHEETAH S | 96 | *0.82* | *0.73* | 0.83 | 0.74 | 0.83 | 0.74 | 0.94 | 0.82 | 0.95 | 0.82 | 0.93 | 0.78 | 0.96 | 0.83 | 0.95 | 0.78 | 0.99 | 0.81 |
|  | 250 | *0.79* | *0.71* | 0.81 | 0.73 | 0.81 | 0.73 | 0.94 | 0.82 | 0.94 | 0.76 | 0.99 | 0.80 | 0.96 | 0.83 | 1.06 | 0.82 | 1.00 | 0.81 |
|  | 500 | *0.71* | *0.66* | 0.74 | 0.68 | 0.76 | 0.70 | 0.94 | 0.82 | 0.81 | 0.68 | 0.97 | 0.78 | 0.96 | 0.83 | 1.10 | 0.82 | 0.95 | 0.78 |
| HOPPER D | 96 | *0.60* | *0.43* | 0.60 | 0.44 | 0.62 | 0.44 | 0.63 | 0.45 | 0.66 | 0.48 | 0.60 | 0.44 | 0.75 | 0.53 | 0.63 | 0.46 | 1.18 | 0.61 |
|  | 250 | *0.65* | *0.46* | 0.67 | 0.48 | 0.69 | 0.50 | 0.70 | 0.49 | 0.80 | 0.56 | 0.69 | 0.50 | 0.77 | 0.54 | 0.74 | 0.54 | 1.28 | 0.68 |
|  | 500 | *0.57* | *0.43* | 0.58 | 0.45 | 0.64 | 0.48 | 0.65 | 0.49 | 0.68 | 0.49 | 0.65 | 0.50 | 0.70 | 0.49 | 0.84 | 0.59 | 1.11 | 0.67 |
| AVG. |  | *0.69* | *0.57* | 0.70 | 0.59 | 0.73 | 0.60 | 0.80 | 0.65 | 0.81 | 0.63 | 0.81 | 0.63 | 0.85 | 0.67 | 0.89 | 0.67 | 1.08 | 0.73 |

Table 6: Benchmark Results on the real-life datasets in the scarce training trajectories regime, *we limited the training sets to only* 1000 *trajectories and have used shorter time limits*. The resulting metrics are shown rounded to two decimal places.

| DATASET | L | N-HITS MSE | MAE | NLINEAR MSE | MAE | LATENT NLINEAR MSE | MAE | SPACETIME MSE | MAE | LSTM MSE | MAE | PATCHTST MSE | MAE | DEEPAR VANILLA MSE | MAE | DEEPAR CL MSE | MAE | NEURAL ODE MSE | MAE |
|---|---|---|---|---|---|---|---|---|---|---|---|---|---|---|---|---|---|---|---|
| ETTM1+2 | 720 | 0.17 | 0.30 | 0.16 | 0.29 | *0.16* | *0.29* | 0.32 | 0.42 | 0.19 | 0.32 | 0.23 | 0.36 | 0.39 | 0.50 | 0.31 | 0.42 | 0.49 | 0.58 |
|  | 336 | 0.19 | 0.32 | 0.19 | 0.32 | *0.19* | *0.32* | 0.19 | 0.32 | 0.22 | 0.34 | 0.26 | 0.37 | 0.34 | 0.44 | 0.44 | 0.47 | 1.50 | 0.93 |
|  | 96 | *0.22* | 0.34 | 0.23 | 0.34 | 0.22 | *0.33* | 0.22 | 0.34 | 0.25 | 0.36 | 0.32 | 0.41 | 0.53 | 0.54 | 0.73 | 0.71 | 0.51 | 0.55 |
| ETTM (LONG) | 1000 | 0.19 | 0.32 | 0.18 | 0.31 | *0.16* | *0.29* | 0.21 | 0.34 | 0.18 | 0.31 | 0.25 | 0.37 | 0.32 | 0.43 | 0.36 | 0.45 | N/A | N/A |
| M4 | 168 | *0.14* | *0.17* | 0.16 | 0.19 | 0.15 | 0.18 | 0.18 | 0.20 | 0.15 | 0.19 | 0.30 | 0.28 | 0.17 | 0.22 | 0.16 | 0.20 | 0.16 | 0.19 |
|  | 96 | *0.24* | *0.24* | 0.26 | 0.25 | 0.25 | 0.25 | 0.26 | 0.27 | 0.25 | 0.26 | 0.30 | 0.28 | 0.37 | 0.34 | 0.30 | 0.29 | 0.25 | 0.25 |
| AVG. |  | *0.15* | 0.23 | 0.16 | 0.23 | 0.16 | *0.23* | 0.18 | 0.25 | 0.19 | 0.25 | 0.26 | 0.29 | 0.29 | 0.34 | 0.33 | 0.35 | 0.45 | 0.40 |

| DATASET | L | LSTM MSE | MAE | LATENT NLINEAR MSE | MAE | SPACETIME MSE | MAE | DEEPAR VANILLA MSE | MAE | DEEPAR CL MSE | MAE | PATCHTST MSE | MAE |
|---|---|---|---|---|---|---|---|---|---|---|---|---|---|
| PEMS_BAY | 144 | *0.64* | *0.36* | 0.68 | 0.40 | 0.68 | 0.39 | 0.72 | 0.38 | 0.74 | 0.38 | 0.87 | 0.45 |

note that the computation time scales significantly with the lookback and history lengths, as much as the dimensionality of the states or the number of estimators used. XGBoost outperformed other tested methods on M4 and was very competitive on ETT. However, chaotic and MuJoCo datasets were underperforming on longer lookback lengths, probably due to the vast number of variables, which rendered the model unable to use them properly (e.g., Hopper dataset results in $500 \cdot 11$ input

Table 7: Benchmark Results on the MuJoCo D datasets described in Sec. 2.1.

| DATASET | L | DEEPAR CL MSE | MAE | LSTM MSE | MAE | LATENT NLINEAR MSE | MAE | SPACETIME MSE | MAE | N-HITS MSE | MAE | NLINEAR MSE | MAE | DEEPAR VANILLA MSE | MAE | NEURAL ODE MSE | MAE |
|---|---|---|---|---|---|---|---|---|---|---|---|---|---|---|---|---|---|
| CHEETAH(D) | 96 | *0.78* | *0.71* | 0.79 | 0.71 | 0.80 | 0.72 | 0.79 | 0.71 | 0.81 | 0.72 | 0.81 | 0.72 | 0.92 | 0.79 | 0.93 | 0.81 |
| | 250 | *0.75* | *0.68* | 0.76 | 0.69 | 0.77 | 0.70 | 0.77 | 0.70 | 0.80 | 0.71 | 0.79 | 0.71 | 0.96 | 0.83 | 0.95 | 0.82 |
| | 500 | *0.65* | *0.62* | 0.66 | 0.63 | 0.69 | 0.64 | 0.68 | 0.64 | 0.75 | 0.68 | 0.72 | 0.66 | 0.94 | 0.82 | 0.91 | 0.80 |
| HOPPER(D) | 96 | 0.58 | 0.42 | 0.56 | *0.41* | 0.58 | 0.42 | 0.59 | 0.43 | *0.56* | 0.41 | 0.61 | 0.43 | 0.58 | 0.42 | 0.63 | 0.47 |
| | 250 | 0.60 | 0.44 | *0.60* | *0.44* | 0.61 | 0.45 | 0.64 | 0.47 | 0.62 | 0.46 | 0.66 | 0.48 | 0.60 | 0.44 | 0.66 | 0.48 |
| | 500 | 0.44 | 0.38 | *0.43* | *0.37* | 0.49 | 0.42 | 0.50 | 0.42 | 0.51 | 0.43 | 0.56 | 0.46 | 0.45 | 0.39 | 0.54 | 0.45 |
| WALKER(D) | 96 | 0.73 | 0.53 | *0.73* | *0.52* | 0.74 | 0.53 | 0.74 | 0.53 | 0.74 | 0.53 | 0.74 | 0.53 | 0.74 | 0.53 | 0.75 | 0.55 |
| | 250 | *0.83* | *0.59* | 0.83 | 0.59 | 0.84 | 0.60 | 0.84 | 0.60 | 0.85 | 0.61 | 0.86 | 0.61 | 0.83 | 0.59 | 0.85 | 0.61 |
| | 500 | *0.79* | *0.55* | 0.80 | 0.56 | 0.89 | 0.62 | 0.87 | 0.62 | 0.93 | 0.64 | 0.96 | 0.67 | 0.82 | 0.56 | 0.86 | 0.60 |
| AVG. | | *0.68* | *0.55* | 0.69 | 0.55 | 0.71 | 0.57 | 0.71 | 0.57 | 0.73 | 0.58 | 0.74 | 0.59 | 0.76 | 0.60 | 0.79 | 0.62 |

variables while having only training 18000 trajectories). The table 8 details the results. All the hyperparameters are kept constant and use their default values, apart from the max depth of the tree that is set to 3, the regression objective is the squared error, the tree method is set to 'gpu_hist' and the number of estimators is varied and shown in Tab. 8.

Table 8: Summary of test MSE for different lookback lengths and datasets using XGBoost estimator

| L | DATASET | MSE | # OF ESTIMATORS |
|---|---|---|---|
| 96 | ETTM1+2 | 0.220 | 32 |
| 336 | ETTM1+2 | 0.176 | 32 |
| 720 | ETTM1+2 | 0.148 | 32 |
| 96 | M4 | 0.167 | 32 |
| 168 | M4 | 0.097 | 32 |
| 96 | M.-G. | 0.724 | 32 |
| 500 | M.-G. | 0.806 | 32 |
| 1000 | M.-G. | 0.811 | 32 |
| 96 | M.-G. | 0.714 | 16 |
| 250 | M.-G. | 0.747 | 16 |
| 500 | M.-G. | 0.710 | 16 |

## A.4 PARAMETER COUNTS

In the paper, we benchmarked representatives of diverse NN model classes whose parameter efficiency varies significantly. We summarize in Tab. 9 the parameter counts of all of the tested models providing insights on how the NN models scale with respect to the state dimension and encoded sequence length.

## A.5 OPEN-SOURCE SOFTWARE DESCRIPTION

Our software library shared in the supplement includes custom implementations of DeepAR, LSTM, LatentODE, NLinear, and Latent NLinear models. We modified the original N-Hits (which we could not distribute due to licensing issues) and SpaceTime implementations to fit our workflows, including support for multivariate series in the N-Hits model. Our framework offers full training and validation workflows, with monitoring and visualizations via ML-Ops (we used neptune.ai). It is designed to be modular, allowing users to swap datasets, trainer configurations, and models easily. We will release the code on GitHub under Apache 2.0 license upon publishing the paper. Instructions are available in the README file within the zipped software package.

Table 9: Parameter counts for the benchmarked NN models. The parameter count numbers are are rounded down and provided in thousands.

| DATASET | MODEL | ENC. LEN. | PARAM. COUNT | ENC. LEN. | PARAM. COUNT | ENC. LEN. | PARAM. COUNT |
|---|---|---|---|---|---|---|---|
| CHEETAH | LATENT ODE | 96 | 190K | 250 | 190K | 500 | 190K |
| CHEETAH | NLINEAR | 96 | 25095K | 250 | 54200K | 500 | 72258K |
| CHEETAH | LATENT NLINEAR | 96 | 34897K | 250 | 75181K | 500 | 100176K |
| CHEETAH | DEEPAR | 96 | 341K | 250 | 341K | 500 | 341K |
| CHEETAH | LSTM | 96 | 420K | 250 | 420K | 500 | 420K |
| CHEETAH | SPACETIME | 96 | 57K | 250 | 57K | 500 | 57K |
| CHEETAH | N-HITS | 96 | 195641K | 250 | 254696K | 500 | 350382K |
| CHEETAH | PATCHTST | 96 | 4913K | 250 | 4921K | 500 | 4934K |
| LORENZ | LATENT ODE | 96 | 177K | 500 | 177K | 1000 | 177K |
| LORENZ | NLINEAR | 96 | 1650K | 500 | 6754K | 1000 | 9003K |
| LORENZ | LATENT NLINEAR | 96 | 18454K | 500 | 75172K | 1000 | 100167K |
| LORENZ | DEEPAR | 96 | 203K | 500 | 203K | 1000 | 203K |
| LORENZ | LSTM | 96 | 407K | 500 | 407K | 1000 | 407K |
| LORENZ | SPACETIME | 96 | 53K | 500 | 53K | 1000 | 53K |
| LORENZ | N-HITS | 96 | 9738K | 500 | 14563K | 1000 | 20537K |
| LORENZ | PATCHTST | 96 | 4769K | 500 | 4790K | 1000 | 4816K |
| K.-S. | LATENT NLINEAR | 96 | 8883K | 250 | 18953K | 500 | 25201K |
| K.-S. | DEEPAR | 96 | 394K | 250 | 394K | 500 | 394K |
| K.-S. | LSTM | 96 | 776K | 250 | 776K | 500 | 776K |
| K.-S. | SPACETIME | 96 | 78K | 250 | 78K | 500 | 78K |
| K.-S. | PATCHTST | 96 | 209817K | 250 | 240505K | 500 | 160192K |
| ETDATASET | LATENT ODE | 96 | 367K | 336 | 367K | 720 | 367K |
| ETDATASET | NLINEAR | 96 | 130K | 336 | 372K | 720 | 519K |
| ETDATASET | LATENT NLINEAR | 96 | 654K | 336 | 1622K | 720 | 2211K |
| ETDATASET | DEEPAR | 96 | 331K | 336 | 331K | 720 | 331K |
| ETDATASET | LSTM | 96 | 662K | 336 | 662K | 720 | 662K |
| ETDATASET | SPACETIME | 96 | 72K | 336 | 72K | 720 | 72K |
| ETDATASET | N-HITS | 96 | 861K | 336 | 1180K | 720 | 1692K |
| ETDATASET | PATCHTST | 96 | 10938K | 336 | 23410K | 720 | 31319K |
| PEMS-BAY | LATENT NLINEAR | 144 | 343537K | | | | |
| PEMS-BAY | DEEPAR | 144 | 1733K | | | | |
| PEMS-BAY | LSTM | 144 | 3382K | | | | |
| PEMS-BAY | SPACETIME | 144 | 454K | | | | |
| PEMS-BAY | PATCHTST | 144 | 212608K | | | | |

## A.6 HYPERPARAMETERS

In Tab. 10 we present hyperparameters for the main dataset classes representatives.

We have tuned the most important hyperparameters of the two introduced models (Latent NLinear & DeepAR + CL). In the case of Latent NLinear, we varied the encoder width and the latent space dimension. We find that the latent NLinear model is pretty robust w.r.t. its hyperparameters. In the case of DeepAR + CL, we varied the model depth and the hidden dimension. The results for DeepAR + CL are generally within the std.dev. margin for the smaller architectures and deteriorate when using larger architectures (hidden dim $> 128$ or depth $> 3$). We provide a simple ablation study in Tab. 11.

## B THE NORMALIZED DATASETS STATISTICS

We report on basic statistics for the studied datasets. Tables 12,13,14,15,16 contain a few quantities describing the component-wise distribution of each dataset. Quantities include the mean, median, and std. dev., min, max values, skewness, kurtosis, 25/50/75th percentiles, and the ratio of outliers found in the dataset (outl.*), the mean, median and std. dev. are also computed without considering the outliers (w/o outl.). Where the number of outliers is counted as the number of points outside the range $[Q_1 - 1.5 * IQR, Q_3 + 1.5 * IQR]$. Where $Q_1$ and $Q_3$ are the 25/75th and $IQR = Q_3 - Q_1$.

To supplement the dataset statistics, for the selected multivariate real-life and synthetic datasets to check if and how the variables are correlated, we compute the Pearson product-moment correlation coefficients $R$ (the normalized covariance matrix $R_{ij} = \frac{C_{ij}}{C_{ii}C_{jj}}$). Correlation coefficient matrices are presented in form of heatmaps in Fig. 2.

Table 10: Table listing the hyperparameters of all of the studied models.

| dataset | model | hyperparameters |
|---|---|---|
| Cheetah | latent ODE | enc: LSTM (2 layers, 50 hidden dim, 18 - out dim), ode: MLP (3 x 100-dim hidden, ELU act.), dec: MLP (4 x 200-dim hidden, ELU act.) |
| Cheetah | NLinear | Linear(Lookback x Horizon) |
| Cheetah | latent NLinear | enc: LSTM (2 layers, 50 hidden dim, 20 - out dim), Linear((Lookback · 20) x (Horizon · 20)), dec: MLP (4 x 200-dim hidden, ELU act.) |
| Cheetah | DeepAR | Enc and Prob. model (the same): LSTM(3 layers, 128 hidden dim), dec: Linear(128, obs. dim) |
| Cheetah | LSTM | Enc: LSTM(3 layers, 100 hidden dim), rnn: LSTM(3 layers, 100 hidden dim), dec: Linear(100, obs. dim) |
| Cheetah | SpaceTime | mostly default from the original repo, see the code for the details |
| Cheetah | N-Hits | shared weights=False, batchnorm=False, ReLU act., 3 stacks, 9 blocks, 2 layers in block, 128 hidden dim, pool kernel size=2, [168, 24, 1] downsample freqs, linear interpolation, dropout=0, max pooling |
| Cheetah | PatchTST | 3 layers, 512 hidden dim, 16 heads, dropout = 0.2, patch len = 10, stride = 10 |
| Lorenz | latent ODE | enc: LSTM (2 layers, 50 hidden dim, 4 - out dim), ode: MLP (3 x 100-dim hidden, ELU act.), dec: MLP (4 x 200-dim hidden, ELU act.) |
| Lorenz | NLinear | Linear((Lookback · obs. dim) x (Horizon · obs. dim)) |
| Lorenz | latent NLinear | enc: LSTM (2 layers, 50 hidden dim, 10 - out dim), Linear((Lookback · 10) x (Horizon · 10)), dec: MLP (4 x 200-dim hidden, ELU act.) |
| Lorenz | DeepAR | Enc and Prob. model (the same): LSTM(3 layers, 100 hidden dim), dec: Linear(100, obs. dim) |
| Lorenz | LSTM | Enc: LSTM(3 layers, 100 hidden dim), rnn: LSTM(3 layers, 100 hidden dim), dec: Linear(100, obs. dim) |
| Lorenz | SpaceTime | mostly default from the original repo, see the code for the details |
| Lorenz | N-Hits | shared weights=False, batchnorm=False, ReLU act., 3 stacks, 9 blocks, 2 layers in block, 128 hidden dim, pool kernel size=2, [168, 24, 1] downsample freqs, linear interpolation, dropout=0, max pooling |
| Lorenz | PatchTST | 3 layers, 512 hidden dim, 16 heads, dropout = 0.2, patch len = 10, stride = 10 |
| K.-S. | latent NLinear | enc: LSTM (2 layers, 50 hidden dim, 40 - out dim), Linear((Lookback · 40) x (Horizon ·40)), dec: MLP (4 x 200-dim hidden, ELU act.) |
| K.-S. | DeepAR | Enc and Prob. model (the same): LSTM(3 layers, 128 hidden dim), dec: Linear(128, obs. dim) |
| K.-S. | LSTM | Enc: LSTM(3 layers, 128 hidden dim), rnn: LSTM(3 layers, 128 hidden dim), dec: Linear(128, obs. dim) |
| K.-S. | SpaceTime | mostly default from the original repo, see the code for the details |
| K.-S. | PatchTST | 3 layers, 256/128/64 hidden dim, 16 heads, dropout = 0.2, patch len = 10, stride = 10 |
| ETDataset | latent ODE | enc: LSTM (2 layers, 128 hidden dim, 16 - out dim), ode: MLP (3 x 128-dim hidden, ELU act.), dec: MLP (4 x 200-dim hidden, ELU act.) |
| ETDataset | NLinear | Linear(Lookback x Horizon) |
| ETDataset | latent NLinear | enc: LSTM (2 layers, 32 hidden dim, 2 - out dim), Linear((Lookback · 2) x (Horizon · 2)), dec: MLP (4 x 200-dim hidden, ELU act.) |
| ETDataset | DeepAR | Enc and Prob. model (the same): LSTM(3 layers, 64 hidden dim), dec: Linear(64, obs. dim) |
| ETDataset | LSTM | Enc: LSTM(3 layers, 128 hidden dim), rnn: LSTM(3 layers, 128 hidden dim), dec: Linear(128, obs. dim) |
| ETDataset | SpaceTime | mostly default from the original repo, see the code for the details |
| ETDataset | N-Hits | shared weights=False, batchnorm=False, ReLU act., 3 stacks, 9 blocks, 2 layers in block, 128 hidden dim, pool kernel size=2, [168, 24, 1] downsample freqs, linear interpolation, dropout=0, max pooling |
| ETDataset | PatchTST | 3 layers, 512 hidden dim, 16 heads, dropout = 0.2, patch len = 10, stride = 10 |
| Pems-Bay | latent NLinear | enc: LSTM (2 layers, 256 hidden dim, 128 - out dim), Linear((Lookback · 128) x (Horizon · 128)), dec: MLP (4 x 200-dim hidden, ELU act.) |
| Pems-Bay | DeepAR | Enc and Prob. model (the same): LSTM(3 layers, 256 hidden dim), dec: Linear(256, obs. dim) |
| Pems-Bay | LSTM | Enc: LSTM(3 layers, 256 hidden dim), rnn: LSTM(3 layers, 256 hidden dim), dec: Linear(256, obs. dim) |
| Pems-Bay | SpaceTime | mostly default from the original repo, see the code for the details |
| Pems-Bay | PatchTST | 3 layers, 320 hidden dim, 16 heads, dropout = 0.2, patch len = 10, stride = 10 |

Table 11: Table listing the hyperparameters that we tuned for the introduced methods (Latent NLinear and DeepAR + CL)

| method | dataset | hyperparameters | MSE | MAE |
|---|---|---|---|---|
| Latent NLinear | ETT | latent dim=2, hidden dim=16 | 0.152 | 0.287 |
| Latent NLinear | ETT | latent dim=2, hidden dim=32 | 0.152 | 0.286 |
| Latent NLinear | ETT | latent dim=2,, hidden dim=64 | 0.151 | 0.286 |
| Latent NLinear | ETT | latent dim=2,, hidden dim=128 | 0.151 | 0.285 |
| Latent NLinear | ETT | latent dim=4, hidden dim=128 | 0.151 | 0.287 |
| Latent NLinear | ETT | latent dim=8, hidden dim=128 | 0.154 | 0.288 |
| Latent NLinear | ETT | latent dim=16, hidden dim=128 | 0.152 | 0.289 |
| DeepAR + CL | ETT | hidden dim=32, depth=3 | 0.177 | 0.308 |
| DeepAR + CL | ETT | hidden dim=64, depth=3 | 0.174 | 0.310 |
| DeepAR + CL | ETT | hidden dim=128, depth=3 | 0.174 | 0.308 |
| DeepAR + CL | ETT | hidden dim=256, depth=3 | 0.411 | 0.487 |
| DeepAR + CL | ETT | hidden dim=128, depth=4 | 0.263 | 0.365 |
| DeepAR + CL | ETT | hidden dim=128, depth=2 | 0.150 | 0.290 |

Table 12: The Basic Dataset Statistics

**M4**

| DIM. | MEAN | MEDIAN | STD DEV | MIN | MAX | SKEWNESS | KURTOSIS | 25TH PER. | 50TH PER. | 75TH PER. | OUTL.* | W/O OUTL. MEAN | MEDIAN | STD DEV |
|---|---|---|---|---|---|---|---|---|---|---|---|---|---|---|
| 1 | 0.000 | -0.322 | 1.000 | -1.102 | 14.221 | 2.183 | 7.403 | -0.675 | -0.322 | 0.361 | 0.055 | -0.168 | -0.370 | 0.684 |

**ETT**

| DIM. | MEAN | MEDIAN | STD DEV | MIN | MAX | SKEWNESS | KURTOSIS | 25TH PER. | 50TH PER. | 75TH PER. | OUTL.* | W/O OUTL. MEAN | MEDIAN | STD DEV |
|---|---|---|---|---|---|---|---|---|---|---|---|---|---|---|
| 1 | 0.000 | -0.124 | 1.000 | -2.168 | 3.557 | 0.777 | 0.390 | -0.773 | -0.124 | 0.558 | 0.020 | -0.059 | -0.156 | 0.920 |

**ELECTRICITY**

| DIM. | MEAN | MEDIAN | STD DEV | MIN | MAX | SKEWNESS | KURTOSIS | 25TH PER. | 50TH PER. | 75TH PER. | OUTL.* | W/O OUTL. MEAN | MEDIAN | STD DEV |
|---|---|---|---|---|---|---|---|---|---|---|---|---|---|---|
| 1 | -0.000 | -0.171 | 1.000 | -0.231 | 36.880 | 19.773 | 516.502 | -0.212 | -0.171 | -0.082 | 0.134 | -0.162 | -0.183 | 0.073 |

**WEATHER**

| DIM. | MEAN | MEDIAN | STD DEV | MIN | MAX | SKEWNESS | KURTOSIS | 25TH PER. | 50TH PER. | 75TH PER. | OUTL.* | W/O OUTL. MEAN | MEDIAN | STD DEV |
|---|---|---|---|---|---|---|---|---|---|---|---|---|---|---|
| 1 | 0.009 | 0.070 | 1.000 | -3.900 | 3.301 | -0.256 | 0.368 | -0.586 | 0.070 | 0.664 | 0.018 | 0.034 | 0.080 | 0.934 |
| 2 | 0.000 | -0.076 | 1.000 | -3.832 | 3.318 | 0.215 | -0.368 | -0.783 | -0.076 | 0.726 | 0.002 | -0.001 | -0.078 | 0.992 |
| 3 | -0.000 | -0.067 | 1.000 | -3.715 | 3.322 | 0.187 | -0.355 | -0.781 | -0.067 | 0.723 | 0.002 | 0.001 | -0.067 | 0.992 |
| 4 | 0.000 | -0.042 | 1.000 | -4.269 | 2.675 | -0.075 | -0.509 | -0.766 | -0.042 | 0.809 | 0.001 | 0.003 | -0.040 | 0.994 |
| 5 | -0.000 | 0.167 | 1.000 | -3.227 | 1.347 | -0.576 | -0.556 | -0.683 | 0.167 | 0.824 | 0.000 | 0.001 | 0.168 | 0.999 |
| 6 | 0.000 | -0.272 | 1.000 | -1.651 | 6.011 | 1.339 | 2.231 | -0.762 | -0.272 | 0.522 | 0.027 | -0.089 | -0.305 | 0.849 |
| 7 | -0.000 | -0.221 | 1.000 | -2.081 | 4.084 | 0.650 | -0.348 | -0.807 | -0.221 | 0.710 | 0.002 | -0.005 | -0.224 | 0.992 |
| 8 | -0.000 | -0.378 | 1.000 | -0.814 | 8.008 | 2.263 | 6.610 | -0.648 | -0.378 | 0.254 | 0.081 | -0.231 | -0.432 | 0.581 |
| 9 | 0.000 | -0.222 | 1.000 | -2.067 | 4.117 | 0.655 | -0.335 | -0.806 | -0.222 | 0.709 | 0.002 | -0.006 | -0.226 | 0.992 |
| 10 | 0.000 | -0.220 | 1.000 | -2.075 | 4.097 | 0.649 | -0.346 | -0.804 | -0.220 | 0.711 | 0.002 | -0.005 | -0.223 | 0.992 |
| 11 | -0.000 | -0.010 | 1.000 | -2.912 | 4.227 | 0.106 | -0.307 | -0.733 | -0.010 | 0.737 | 0.002 | -0.008 | -0.014 | 0.987 |
| 12 | -0.000 | -0.013 | 1.000 | -395.635 | 0.482 | -393.967 | 155801.447 | -0.042 | -0.013 | 0.029 | 0.044 | -0.006 | -0.017 | 0.047 |
| 13 | -0.000 | -0.256 | 1.000 | -1.466 | 8.062 | 1.492 | 3.054 | -0.734 | -0.256 | 0.468 | 0.036 | -0.116 | -0.297 | 0.802 |
| 14 | 0.000 | 0.244 | 1.000 | -2.191 | 2.210 | -0.562 | -0.346 | -0.443 | 0.244 | 0.654 | 0.012 | 0.026 | 0.254 | 0.978 |
| 15 | -0.000 | -0.080 | 1.000 | -0.080 | 151.926 | 63.343 | 6872.838 | -0.080 | -0.080 | -0.080 | 0.033 | -0.080 | -0.080 | 0.000 |
| 16 | -0.000 | -0.254 | 1.000 | -0.254 | 4.702 | 4.032 | 14.979 | -0.254 | -0.254 | -0.254 | 0.077 | -0.254 | -0.254 | 0.000 |
| 17 | 0.000 | -0.591 | 1.001 | -0.607 | 4.713 | 1.856 | 2.638 | -0.607 | -0.591 | 0.230 | 0.109 | -0.295 | -0.607 | 0.521 |
| 18 | 0.000 | -0.589 | 1.000 | -0.612 | 4.681 | 1.858 | 2.694 | -0.612 | -0.589 | 0.246 | 0.104 | -0.285 | -0.612 | 0.536 |
| 19 | -0.000 | -0.539 | 1.000 | -19.068 | 4.064 | -0.804 | 45.998 | -0.563 | -0.539 | 0.212 | 0.114 | -0.279 | -0.563 | 0.474 |
| 20 | -0.000 | -0.132 | 1.000 | -3.316 | 3.383 | 0.442 | -0.260 | -0.779 | -0.132 | 0.676 | 0.003 | -0.007 | -0.135 | 0.986 |
| 21 | -0.000 | 0.035 | 1.000 | -22.842 | 0.303 | -22.754 | 516.614 | 0.016 | 0.035 | 0.064 | 0.035 | 0.040 | 0.033 | 0.034 |

Table 13: The Basic Dataset Statistics cont.

| | PEMS-BAY (FIRST 20) | | | | | | | | | | | W/O OUTL. | | |
|---|---|---|---|---|---|---|---|---|---|---|---|---|---|---|
| Dim. | Mean | Median | Std Dev | Min | Max | Skewness | Kurtosis | 25th Per. | 50th Per. | 75th Per. | Outl.* | Mean | Median | Std Dev |
| 1 | -0.000 | 0.334 | 1.000 | -6.918 | 1.534 | -3.636 | 15.829 | -0.009 | 0.334 | 0.466 | 0.129 | 0.297 | 0.374 | 0.293 |
| 2 | -0.000 | 0.366 | 1.000 | -3.684 | 0.962 | -2.381 | 4.131 | 0.225 | 0.366 | 0.444 | 0.142 | 0.375 | 0.389 | 0.135 |
| 3 | -0.000 | 0.307 | 1.000 | -4.928 | 1.622 | -2.481 | 7.226 | -0.293 | 0.307 | 0.633 | 0.063 | 0.209 | 0.386 | 0.548 |
| 4 | -0.000 | 0.024 | 1.000 | -5.623 | 1.430 | -1.930 | 4.349 | -0.291 | 0.024 | 0.763 | 0.064 | 0.207 | 0.157 | 0.609 |
| 5 | -0.000 | 0.227 | 1.000 | -5.244 | 1.973 | -1.614 | 2.976 | -0.318 | 0.227 | 0.580 | 0.080 | 0.235 | 0.247 | 0.603 |
| 6 | -0.000 | 0.274 | 1.000 | -5.933 | 0.845 | -3.847 | 14.500 | 0.097 | 0.274 | 0.383 | 0.097 | 0.264 | 0.294 | 0.190 |
| 7 | 0.000 | 0.349 | 1.000 | -4.235 | 0.969 | -2.404 | 5.101 | -0.012 | 0.349 | 0.569 | 0.114 | 0.324 | 0.427 | 0.325 |
| 8 | -0.000 | 0.329 | 1.000 | -7.208 | 1.892 | -2.885 | 9.288 | 0.054 | 0.329 | 0.467 | 0.145 | 0.321 | 0.379 | 0.246 |
| 9 | -0.000 | 0.332 | 1.000 | -4.950 | 1.211 | -2.717 | 6.433 | 0.166 | 0.332 | 0.453 | 0.149 | 0.344 | 0.369 | 0.174 |
| 10 | -0.000 | 0.409 | 1.000 | -2.882 | 0.789 | -1.985 | 2.246 | 0.182 | 0.409 | 0.528 | 0.152 | 0.406 | 0.460 | 0.177 |
| 11 | 0.000 | 0.230 | 1.000 | -7.994 | 1.092 | -5.758 | 37.845 | -0.103 | 0.230 | 0.367 | 0.050 | 0.165 | 0.250 | 0.338 |
| 12 | -0.000 | 0.366 | 1.000 | -4.609 | 1.439 | -2.589 | 5.529 | 0.163 | 0.366 | 0.463 | 0.138 | 0.355 | 0.398 | 0.170 |
| 13 | 0.000 | 0.209 | 1.000 | -7.725 | 2.375 | -2.451 | 10.523 | -0.219 | 0.209 | 0.461 | 0.103 | 0.218 | 0.234 | 0.539 |
| 14 | 0.000 | 0.136 | 1.000 | -13.280 | 3.659 | -2.845 | 21.933 | -0.433 | 0.136 | 0.599 | 0.029 | 0.097 | 0.136 | 0.751 |
| 15 | 0.000 | 0.271 | 1.000 | -8.591 | 2.773 | -3.307 | 16.138 | -0.078 | 0.271 | 0.435 | 0.105 | 0.207 | 0.291 | 0.344 |
| 16 | 0.000 | 0.366 | 1.000 | -4.541 | 1.156 | -2.596 | 5.453 | 0.216 | 0.366 | 0.441 | 0.152 | 0.365 | 0.382 | 0.130 |
| 17 | 0.000 | 0.154 | 1.000 | -13.249 | 2.148 | -6.294 | 60.119 | -0.220 | 0.154 | 0.465 | 0.052 | 0.139 | 0.185 | 0.463 |
| 18 | 0.000 | 0.295 | 1.000 | -3.867 | 1.378 | -1.812 | 2.491 | -0.114 | 0.295 | 0.576 | 0.124 | 0.338 | 0.343 | 0.420 |
| 19 | 0.000 | 0.349 | 1.000 | -5.786 | 0.699 | -3.478 | 13.459 | -0.142 | 0.349 | 0.506 | 0.074 | 0.242 | 0.384 | 0.356 |
| 20 | 0.000 | 0.300 | 1.000 | -4.236 | 1.200 | -1.783 | 2.549 | -0.237 | 0.300 | 0.713 | 0.093 | 0.274 | 0.440 | 0.531 |

Table 14: The Basic Dataset Statistics cont.

| | | | | | | MSO | | | | | | | W/O OUTL. | |
| DIM. | MEAN | MEDIAN | STD DEV | MIN | MAX | SKEWNESS | KURTOSIS | 25TH PER. | 50TH PER. | 75TH PER. | OUTL.* | MEAN | MEDIAN | STD DEV |
|---|---|---|---|---|---|---|---|---|---|---|---|---|---|---|
| 1 | 0.000 | -0.002 | 1.000 | -1.999 | 1.995 | -0.001 | -0.754 | -0.721 | -0.002 | 0.725 | 0.000 | 0.000 | -0.002 | 1.000 |

| | | | | | | LOTKA | | | | | | | W/O OUTL. | |
| DIM. | MEAN | MEDIAN | STD DEV | MIN | MAX | SKEWNESS | KURTOSIS | 25TH PER. | 50TH PER. | 75TH PER. | OUTL.* | MEAN | MEDIAN | STD DEV |
|---|---|---|---|---|---|---|---|---|---|---|---|---|---|---|
| 1 | -0.000 | -0.496 | 1.000 | -0.658 | 5.136 | 1.987 | 3.609 | -0.622 | -0.496 | 0.189 | 0.111 | -0.298 | -0.539 | 0.504 |
| 2 | 0.000 | -0.441 | 1.000 | -0.928 | 4.707 | 1.439 | 1.562 | -0.743 | -0.441 | 0.499 | 0.034 | -0.105 | -0.475 | 0.839 |

| | | | | | | MACKEY-GLASS | | | | | | | W/O OUTL. | |
| DIM. | MEAN | MEDIAN | STD DEV | MIN | MAX | SKEWNESS | KURTOSIS | 25TH PER. | 50TH PER. | 75TH PER. | OUTL.* | MEAN | MEDIAN | STD DEV |
|---|---|---|---|---|---|---|---|---|---|---|---|---|---|---|
| 1 | -0.000 | 0.141 | 1.000 | -2.243 | 1.835 | -0.404 | -0.724 | -0.665 | 0.141 | 0.753 | 0.000 | -0.000 | 0.141 | 1.000 |

| | | | | | | LORENZ | | | | | | | W/O OUTL. | |
| DIM. | MEAN | MEDIAN | STD DEV | MIN | MAX | SKEWNESS | KURTOSIS | 25TH PER. | 50TH PER. | 75TH PER. | OUTL.* | MEAN | MEDIAN | STD DEV |
|---|---|---|---|---|---|---|---|---|---|---|---|---|---|---|
| 1 | 0.000 | 0.035 | 1.000 | -2.591 | 2.280 | -0.110 | -1.008 | -0.885 | 0.035 | 0.886 | 0.000 | 0.000 | 0.035 | 1.000 |
| 2 | 0.000 | 0.018 | 1.000 | -3.124 | 2.691 | -0.117 | -0.551 | -0.804 | 0.018 | 0.811 | 0.000 | 0.000 | 0.018 | 1.000 |
| 3 | -0.000 | 0.086 | 1.000 | -2.828 | 3.443 | -0.084 | 0.176 | -0.640 | 0.086 | 0.600 | 0.018 | 0.011 | 0.091 | 0.939 |

| | | | | | | CHEETAH (STOCHASTIC) | | | | | | | W/O OUTL. | |
| DIM. | MEAN | MEDIAN | STD DEV | MIN | MAX | SKEWNESS | KURTOSIS | 25TH PER. | 50TH PER. | 75TH PER. | OUTL.* | MEAN | MEDIAN | STD DEV |
|---|---|---|---|---|---|---|---|---|---|---|---|---|---|---|
| 1 | 0.000 | -0.074 | 1.000 | -3.302 | 6.131 | 1.023 | 3.873 | -0.589 | -0.074 | 0.484 | 0.041 | -0.077 | -0.097 | 0.783 |
| 2 | -0.000 | 0.019 | 1.000 | -5.059 | 3.280 | -0.391 | 1.224 | -0.577 | 0.019 | 0.629 | 0.024 | 0.043 | 0.033 | 0.891 |
| 3 | 0.000 | 0.017 | 1.000 | -2.146 | 1.964 | -0.066 | -1.165 | -0.858 | 0.017 | 0.885 | 0.000 | 0.000 | 0.017 | 1.000 |
| 4 | 0.000 | 0.034 | 1.000 | -2.326 | 1.769 | -0.090 | -1.278 | -0.908 | 0.034 | 0.930 | 0.000 | 0.000 | 0.034 | 1.000 |
| 5 | -0.000 | 0.013 | 1.000 | -1.965 | 1.934 | -0.027 | -1.293 | -0.914 | 0.013 | 0.914 | 0.000 | -0.000 | 0.013 | 1.000 |
| 6 | -0.000 | 0.011 | 1.000 | -2.502 | 2.891 | 0.023 | -0.996 | -0.856 | 0.011 | 0.845 | 0.000 | -0.000 | 0.011 | 1.000 |
| 7 | 0.000 | -0.129 | 1.000 | -2.557 | 3.861 | 0.444 | -0.074 | -0.712 | -0.129 | 0.658 | 0.007 | -0.020 | -0.138 | 0.972 |
| 8 | -0.000 | -0.141 | 1.000 | -1.961 | 6.565 | 1.762 | 5.434 | -0.711 | -0.141 | 0.409 | 0.040 | -0.133 | -0.179 | 0.745 |
| 9 | 0.000 | -0.015 | 1.000 | -3.190 | 2.418 | -0.164 | -0.769 | -0.822 | -0.015 | 0.883 | 0.000 | 0.000 | -0.015 | 1.000 |
| 10 | -0.000 | 0.004 | 1.000 | -3.608 | 3.412 | -0.128 | -0.405 | -0.748 | 0.004 | 0.796 | 0.001 | 0.002 | 0.005 | 0.995 |
| 11 | -0.000 | -0.119 | 1.000 | -3.976 | 3.325 | 0.262 | -0.342 | -0.726 | -0.119 | 0.708 | 0.002 | 0.002 | -0.118 | 0.990 |
| 12 | 0.000 | -0.007 | 1.000 | -1.607 | 1.705 | 0.027 | -1.573 | -1.030 | -0.007 | 1.029 | 0.000 | 0.000 | -0.007 | 1.000 |
| 13 | 0.000 | 0.096 | 1.000 | -1.740 | 1.798 | -0.094 | -1.497 | -0.976 | 0.096 | 0.966 | 0.000 | 0.000 | 0.096 | 1.000 |
| 14 | -0.000 | 0.110 | 1.000 | -1.752 | 1.581 | -0.137 | -1.518 | -1.025 | 0.110 | 0.979 | 0.000 | -0.000 | 0.110 | 1.000 |
| 15 | 0.000 | 0.014 | 1.000 | -2.012 | 2.238 | 0.037 | -1.195 | -0.908 | 0.014 | 0.857 | 0.000 | 0.000 | 0.014 | 1.000 |
| 16 | -0.000 | -0.096 | 1.000 | -3.010 | 3.592 | 0.161 | -0.666 | -0.794 | -0.096 | 0.813 | 0.000 | -0.001 | -0.097 | 0.999 |
| 17 | -0.000 | 0.164 | 1.000 | -4.962 | 3.775 | -0.701 | 1.274 | -0.491 | 0.164 | 0.612 | 0.050 | 0.091 | 0.197 | 0.811 |

Table 15: The Basic Dataset Statistics cont.

| | HOPPER (DETER) | | | | | | | | | | | W/O OUTL. | | |
|---|---|---|---|---|---|---|---|---|---|---|---|---|---|---|
| DIM. | MEAN | MEDIAN | STD DEV | MIN | MAX | SKEWNESS | KURTOSIS | 25TH PER. | 50TH PER. | 75TH PER. | OUTL.* | MEAN | MEDIAN | STD DEV |
| 1 | 0.000 | -0.559 | 1.000 | -0.646 | 2.904 | 1.408 | 0.228 | -0.565 | -0.559 | -0.348 | 0.234 | -0.538 | -0.561 | 0.063 |
| 2 | -0.000 | 0.123 | 1.000 | -4.999 | 2.942 | -0.659 | 0.306 | -0.700 | 0.123 | 0.710 | 0.006 | 0.020 | 0.129 | 0.966 |
| 3 | 0.000 | -0.397 | 1.000 | -2.341 | 2.678 | 1.260 | 0.234 | -0.544 | -0.397 | -0.224 | 0.270 | -0.468 | -0.438 | 0.178 |
| 4 | -0.000 | 0.280 | 1.000 | -7.470 | 1.239 | -4.390 | 20.830 | 0.264 | 0.280 | 0.286 | 0.320 | 0.280 | 0.281 | 0.011 |
| 5 | 0.000 | 0.130 | 1.000 | -1.839 | 1.684 | -0.189 | -1.175 | -0.904 | 0.130 | 0.779 | 0.000 | 0.000 | 0.130 | 1.000 |
| 6 | -0.000 | -0.277 | 1.000 | -2.254 | 2.785 | 0.407 | -1.025 | -0.851 | -0.277 | 0.937 | 0.000 | -0.000 | -0.277 | 1.000 |
| 7 | 0.000 | 0.097 | 1.000 | -6.346 | 3.744 | -1.452 | 7.880 | -0.052 | 0.097 | 0.263 | 0.259 | 0.115 | 0.102 | 0.213 |
| 8 | 0.000 | -0.004 | 1.000 | -4.851 | 4.839 | -0.354 | 4.064 | -0.250 | -0.004 | 0.329 | 0.157 | 0.034 | -0.000 | 0.400 |
| 9 | 0.000 | 0.024 | 1.000 | -4.372 | 4.497 | 0.110 | 3.282 | -0.299 | 0.024 | 0.311 | 0.178 | 0.009 | 0.030 | 0.416 |
| 10 | -0.000 | -0.001 | 1.000 | -5.653 | 8.806 | 2.140 | 18.873 | -0.060 | -0.001 | 0.009 | 0.298 | -0.012 | 0.000 | 0.038 |
| 11 | 0.000 | 0.090 | 1.000 | -1.786 | 1.775 | -0.138 | -0.630 | -0.583 | 0.090 | 0.601 | 0.000 | -0.000 | 0.090 | 1.000 |
| | KS (FIRST 20) | | | | | | | | | | | W/O OUTL. | | |
| DIM. | MEAN | MEDIAN | STD DEV | MIN | MAX | SKEWNESS | KURTOSIS | 25TH PER. | 50TH PER. | 75TH PER. | OUTL.* | MEAN | MEDIAN | STD DEV |
| 1 | 0.000 | -0.000 | 1.000 | -2.373 | 2.636 | 0.001 | -0.831 | -0.777 | -0.000 | 0.763 | 0.000 | 0.000 | -0.000 | 1.000 |
| 2 | 0.000 | 0.010 | 1.000 | -2.299 | 2.391 | 0.046 | -0.876 | -0.797 | 0.010 | 0.753 | 0.000 | 0.000 | 0.010 | 1.000 |
| 3 | -0.000 | -0.010 | 1.000 | -2.504 | 2.385 | -0.007 | -0.845 | -0.746 | -0.010 | 0.781 | 0.000 | 0.000 | -0.010 | 1.000 |
| 4 | 0.000 | -0.010 | 1.000 | -2.445 | 2.481 | -0.044 | -0.812 | -0.740 | -0.010 | 0.764 | 0.000 | -0.000 | -0.010 | 1.000 |
| 5 | -0.000 | 0.012 | 1.000 | -2.234 | 2.368 | -0.070 | -0.893 | -0.772 | 0.012 | 0.779 | 0.000 | -0.000 | 0.012 | 1.000 |
| 6 | -0.000 | -0.004 | 1.000 | -2.345 | 2.370 | -0.040 | -0.999 | -0.811 | -0.004 | 0.829 | 0.000 | 0.000 | -0.004 | 1.000 |
| 7 | 0.000 | -0.040 | 1.000 | -2.138 | 2.225 | 0.123 | -0.914 | -0.797 | -0.040 | 0.786 | 0.000 | 0.000 | -0.040 | 1.000 |
| 8 | -0.000 | -0.047 | 1.000 | -2.248 | 2.504 | 0.141 | -0.808 | -0.758 | -0.047 | 0.745 | 0.000 | -0.000 | -0.047 | 1.000 |
| 9 | -0.000 | -0.022 | 1.000 | -2.335 | 2.555 | 0.137 | -0.821 | -0.792 | -0.022 | 0.713 | 0.000 | 0.000 | -0.022 | 1.000 |
| 10 | -0.000 | 0.013 | 1.000 | -2.149 | 2.309 | 0.023 | -0.822 | -0.769 | 0.013 | 0.722 | 0.000 | 0.000 | 0.013 | 1.000 |
| 11 | -0.000 | 0.019 | 1.000 | -2.740 | 2.443 | -0.056 | -0.855 | -0.759 | 0.019 | 0.767 | 0.000 | 0.000 | 0.019 | 1.000 |
| 12 | 0.000 | -0.005 | 1.000 | -2.490 | 2.216 | -0.073 | -0.891 | -0.764 | -0.005 | 0.797 | 0.000 | 0.000 | -0.005 | 1.000 |
| 13 | -0.000 | 0.021 | 1.000 | -2.519 | 2.220 | -0.096 | -0.811 | -0.729 | 0.021 | 0.756 | 0.000 | 0.000 | 0.021 | 1.000 |
| 14 | -0.000 | -0.016 | 1.000 | -2.430 | 2.299 | 0.040 | -0.858 | -0.764 | -0.016 | 0.745 | 0.000 | 0.000 | -0.016 | 1.000 |
| 15 | 0.000 | -0.004 | 1.000 | -2.304 | 2.362 | 0.024 | -0.913 | -0.786 | -0.004 | 0.763 | 0.000 | 0.000 | -0.004 | 1.000 |
| 16 | -0.000 | -0.005 | 1.000 | -2.328 | 2.217 | 0.024 | -0.915 | -0.797 | -0.005 | 0.757 | 0.000 | 0.000 | -0.005 | 1.000 |
| 17 | 0.000 | -0.025 | 1.000 | -2.512 | 2.384 | -0.024 | -0.911 | -0.775 | -0.025 | 0.785 | 0.000 | -0.000 | -0.025 | 1.000 |
| 18 | -0.000 | -0.001 | 1.000 | -2.516 | 2.406 | -0.006 | -0.828 | -0.756 | -0.001 | 0.753 | 0.000 | -0.000 | -0.001 | 1.000 |
| 19 | -0.000 | 0.043 | 1.000 | -2.374 | 2.457 | -0.064 | -0.835 | -0.788 | 0.043 | 0.750 | 0.000 | -0.000 | 0.043 | 1.000 |
| 20 | 0.000 | -0.023 | 1.000 | -2.326 | 2.499 | 0.045 | -0.837 | -0.789 | -0.023 | 0.773 | 0.000 | 0.000 | -0.023 | 1.000 |

Table 16: The Basic Dataset Statistics cont.

| | | | | | | | | | | | | | | |
|---|---|---|---|---|---|---|---|---|---|---|---|---|---|---|
| | | | **CAHN-HILLARD** | | | | | | | | | | **W/O OUTL.** | |
| DIM. | MEAN | MEDIAN | STD DEV | MIN | MAX | SKEWNESS | KURTOSIS | 25TH PER. | 50TH PER. | 75TH PER. | OUTL.* | MEAN | MEDIAN | STD DEV |
| 1 | -0.000 | 0.032 | 1.000 | -1.811 | 1.684 | -0.073 | -1.246 | -0.900 | 0.032 | 0.950 | 0.000 | -0.000 | 0.032 | 1.000 |
| 2 | 0.000 | 0.042 | 1.000 | -1.536 | 1.954 | -0.047 | -1.259 | -0.880 | 0.042 | 0.950 | 0.000 | 0.000 | 0.042 | 1.000 |
| 3 | 0.000 | -0.269 | 1.000 | -1.280 | 2.197 | 0.701 | -0.621 | -0.857 | -0.269 | 0.692 | 0.000 | 0.000 | -0.269 | 1.000 |
| 4 | 0.000 | 0.310 | 1.000 | -2.158 | 1.211 | -0.691 | -0.792 | -0.707 | 0.310 | 0.854 | 0.000 | 0.000 | 0.310 | 1.000 |
| 5 | -0.000 | 0.053 | 1.000 | -1.184 | 1.691 | 0.416 | -1.217 | -0.861 | 0.053 | 0.618 | 0.000 | -0.000 | 0.053 | 1.000 |
| 6 | 0.000 | -0.290 | 1.000 | -1.330 | 1.573 | 0.164 | -1.563 | -0.974 | -0.290 | 1.016 | 0.000 | -0.000 | -0.290 | 1.000 |
| 7 | -0.000 | -0.121 | 1.000 | -1.512 | 1.689 | 0.052 | -1.363 | -0.923 | -0.121 | 0.882 | 0.000 | 0.000 | -0.121 | 1.000 |
| 8 | 0.000 | 0.090 | 1.000 | -2.618 | 1.227 | -0.632 | -0.313 | -0.695 | 0.090 | 1.007 | 0.000 | 0.000 | 0.090 | 1.000 |
| 9 | 0.000 | -0.095 | 1.000 | -1.368 | 1.439 | 0.094 | -1.543 | -1.027 | -0.095 | 1.107 | 0.000 | -0.000 | -0.095 | 1.000 |
| 10 | 0.000 | 0.431 | 1.000 | -1.507 | 1.490 | -0.233 | -1.523 | -1.142 | 0.431 | 0.908 | 0.000 | 0.000 | 0.431 | 1.000 |
| 11 | 0.000 | -0.169 | 1.000 | -1.410 | 1.650 | 0.319 | -1.204 | -0.779 | -0.169 | 0.967 | 0.000 | 0.000 | -0.169 | 1.000 |
| 12 | 0.000 | 0.138 | 1.000 | -1.465 | 1.500 | -0.058 | -1.475 | -1.104 | 0.138 | 1.005 | 0.000 | -0.000 | 0.138 | 1.000 |
| 13 | -0.000 | -0.224 | 1.000 | -1.421 | 2.170 | 0.464 | -0.914 | -0.830 | -0.224 | 0.814 | 0.000 | -0.000 | -0.224 | 1.000 |
| 14 | 0.000 | 0.158 | 1.000 | -2.068 | 1.229 | -0.544 | -0.852 | -0.666 | 0.158 | 0.982 | 0.000 | 0.000 | 0.158 | 1.000 |
| 15 | 0.000 | -0.412 | 1.000 | -1.180 | 1.651 | 0.436 | -1.464 | -0.889 | -0.412 | 1.116 | 0.000 | 0.000 | -0.412 | 1.000 |
| 16 | -0.000 | 0.092 | 1.000 | -1.645 | 1.646 | -0.154 | -1.236 | -0.964 | 0.092 | 0.884 | 0.000 | -0.000 | 0.092 | 1.000 |
| 17 | 0.000 | 0.068 | 1.000 | -1.599 | 1.408 | -0.160 | -1.355 | -0.940 | 0.068 | 1.042 | 0.000 | -0.000 | 0.068 | 1.000 |
| 18 | 0.000 | -0.302 | 1.000 | -1.305 | 1.587 | 0.269 | -1.431 | -0.924 | -0.302 | 0.989 | 0.000 | 0.000 | -0.302 | 1.000 |
| 19 | 0.000 | 0.455 | 1.000 | -1.637 | 1.224 | -0.298 | -1.574 | -1.020 | 0.455 | 0.927 | 0.000 | -0.000 | 0.455 | 1.000 |
| 20 | -0.000 | 0.007 | 1.000 | -1.414 | 1.741 | 0.198 | -1.310 | -1.003 | 0.007 | 0.899 | 0.000 | 0.000 | 0.007 | 1.000 |

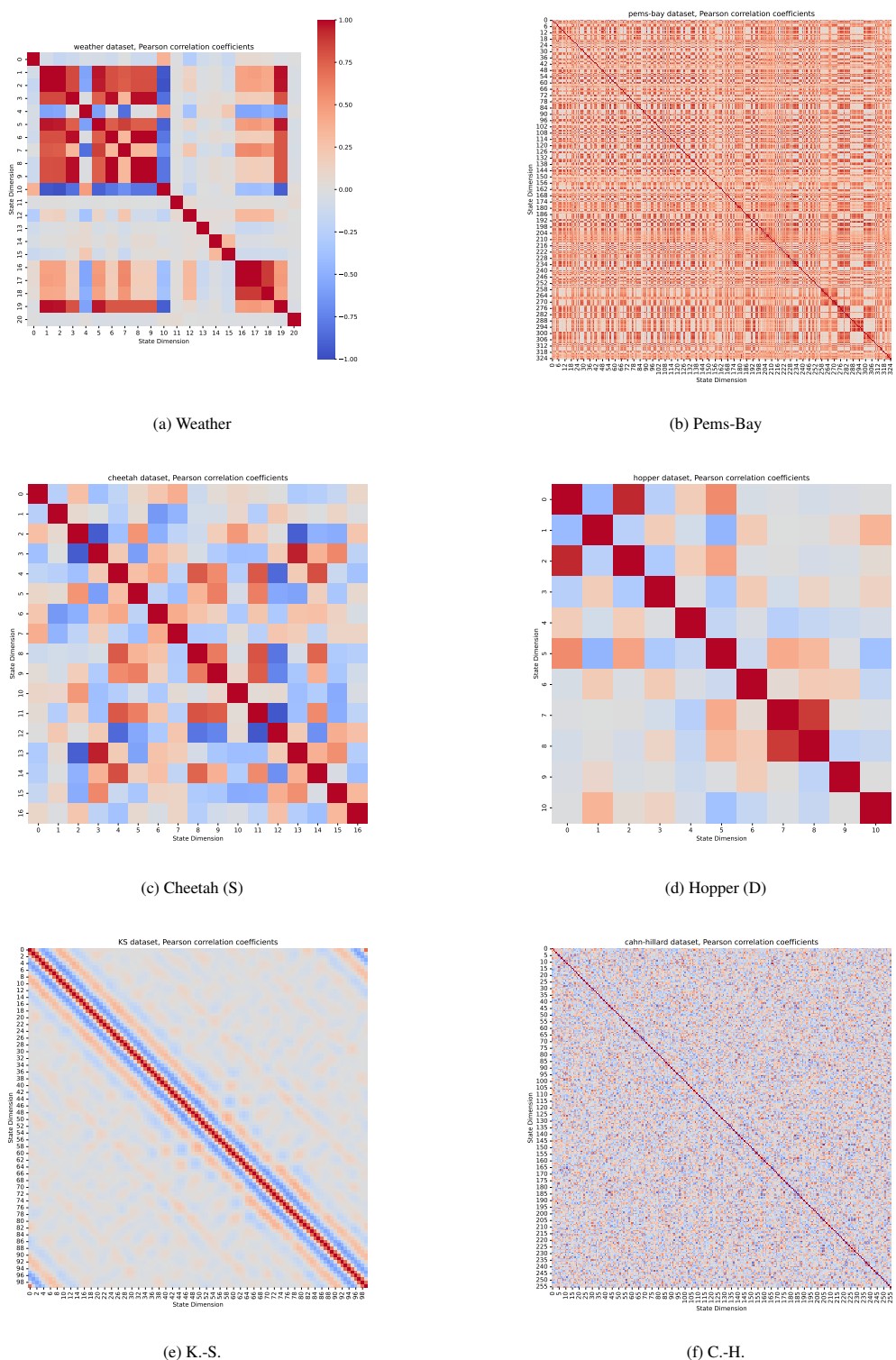

(a) Weather

(b) Pems-Bay

(c) Cheetah (S)

(d) Hopper (D)

(e) K.-S.

(f) C.-H.

Figure 2: Heatmaps of Pearson correlation coefficient matrices presented for the selected benchmark datasets. Fig. 2a,2b presents real-life datasets, whereas Fig. 2c,2d,2e,2f presents synthetic.

