# OpenReview forum: "Unified Long-Term Time-Series Forecasting  Benchmark"
_ICLR.cc/2024/Conference — Submitted to ICLR 2024_

### Official Review · Reviewer_ptT1 · 2023-10-31

**Soundness:** 2 fair
**Presentation:** 3 good
**Contribution:** 2 fair
**Rating:** 5
**Confidence:** 4

**Summary:**

The paper presents a unified benchmark dataset for long-term time-series forecasting (LTSF), addressing the limitations of existing datasets. The dataset includes real-life and synthetic data from diverse domains, enabling comprehensive evaluations of LTSF methods. The datasets are split into training and testing trajectories with fixed lengths, allowing for standardized evaluations. The paper introduces two new hand-crafted models, the latent NLinear model and DeepAR enhanced with curriculum learning, which outperform existing models. The benchmark includes classical and state-of-the-art models such as LSTM, DeepAR, N-Hits, PatchTST, and LatentODE, and evaluates their performance on the dataset. The paper emphasizes the importance of dataset diversity, facilitating ML model training and testing, and introducing new models to improve LTSF accuracy. The paper concludes by providing an open-source library with implementations to promote further advancements in LTSF research.

**Strengths:**

1. Comprehensive dataset: The article presents a comprehensive dataset that incorporates real-life and synthetic data from diverse domains, enabling the evaluation of long-term time-series forecasting methods in a wide range of contexts. This comprehensive dataset helps in gaining a more holistic understanding of the strengths and weaknesses of different methods across various domains.

2. Standardized evaluation: To ensure consistency and comparability, the authors split the dataset into training and testing trajectories with fixed lengths. This standardized evaluation approach allows for more accurate and reliable comparisons between different methods.

3. Introduction of new models: The article introduces two new hand-crafted models, namely the latent NLinear model and DeepAR enhanced with curriculum learning. These new models demonstrate significant improvements across the entire dataset, showcasing their effectiveness and potential in long-term time-series forecasting.

4. Extensive method comparison: The article conducts a thorough benchmarking analysis, evaluating a range of neural network-based models including classical approaches and state-of-the-art methods. By comparing the performance of these methods on the dataset, a better understanding of their strengths and limitations can be gained, fostering advancements in the field.

5. Open-source code library: To facilitate further advancements in the field, the authors provide an open-source code library that includes implementations of the methods discussed. This enables other researchers to easily replicate and extend these methods, accelerating progress in the field.

**Weaknesses:**

1. Regarding innovation:

   (1) The proposed methods in this article seem to have had little impact on the overall benchmark and did not address any gaps in the existing methodological framework or provide substantial insights or innovations. The article needs to clearly articulate the innovative aspects of its methods and how they improve upon existing approaches.

   (2) The benchmark dataset created in the article consists of two parts: SYNTHETIC and REAL-LIFE. However, while the REAL-LIFE part mainly comprises existing time-series datasets, the contribution of the SYNTHETIC part in addressing the limitations of the existing REAL-LIFE datasets is not adequately explained. The article should analyze the potential shortcomings of the existing real-life datasets and explain how the SYNTHETIC dataset complements them.

2. Regarding experiments:

   (1) The description of the process for jointly evaluating models using artificial and real-life data is overly concise and fails to analyze the fundamental differences between artificial and real-life data. It is recommended to propose a systematic evaluation framework for time-series models that goes beyond presenting a series of datasets.

   (2) The experimental evaluation of Transformer models only includes one model, PatchTST, thereby lacking a comprehensive exploration of other Transformer models. It is advisable to include a wider range of Transformer models in the experiments for a more comprehensive comparison and evaluation.

3. Regarding presentation:

   (1) The "BENCHMARK SYNERGY" section lacks formulas and illustrations, making it difficult for readers to understand its content. It is recommended to provide more visualizations and illustrations in this section to aid reader comprehension.

   (2) The article lacks basic visual analysis and conclusions, making it challenging for readers to grasp the experimental results intuitively. Including more data visualizations and clear conclusions in the article would provide more intuitive information.

   (3) The article lacks intuitive descriptions of the datasets, such as data-time graph representations, making it difficult for readers to understand the characteristics and structure of the datasets clearly.

   (4) There is a lack of illustrations for different data generation methods, impeding readers' understanding of the data generation process and methods. It would be beneficial to include relevant figures to help readers comprehend the data generation process.

   (5) The absence of data-time graph representations for the predictive performance of different models hinders readers' ability to visually compare the performance of various models. It is recommended to provide data-time graph representations in the results section to facilitate a better understanding of the predictive performance of the different models.

**Questions:**

1. Flaw: The author says: "The most dominant class of LTSF benchmarks relies on datasets with rather uniform characteristics composed of nine widely-used real-world datasets: Electricity Transformer Temperature (ETT) from Zhou et al. (2021a) (split into four cases: ETTh1, ETTh2, ETTm1, ETTm2), Traffic, Electricity, Weather, ILI, ExchangeRate. All of them are univariate time series with a significant degree of non-deterministicity (these are real-life measurements)." However, they are mostly multivariate time series.

2. What I am concerned about most is listed in the weakness, I won't refuse to raise my points if the author can address my concerns.

---

> ### Author Response · Authors · 2023-11-21
> **Answer to Reviewer ptT1**
>
> We thank the reviewer for the insightful remarks. We address the raised concerns and questions below. We will include the additional clarifications in the revised version of our paper.
>
> >Regarding innovation:
>
> >  (1) The proposed methods in this article seem to have had little impact on the overall benchmark and did not address any gaps in the existing methodological framework or provide substantial insights or innovations...
>
>
> As the paper's main contribution, we consider the introduction of the new synthetic datasets complementing the typically used real-life datasets' comprehensive evaluation of the LTSF ML models on the proposed unified benchmark, giving rise to some interesting conclusions about new models vs. established models' performance.  We propose and evaluate two new model architectures based on existing approaches for LTSF, namely the latent NLinear method and DeepAR with the curriculum learning phase. As demonstrated clearly in the presented benchmark, we observed significant improvements across the entire dataset when using these models, suggesting their effectiveness as baselines for LTSF.
>
> Moreover, the introduced latent NLinear model allows for treating datasets with many variables, thanks to the possibility of using latent spaces having a compacted dimension. Such latent space reduces the number of state variables and can encapture the nonlinear dependencies between different variables in the dataset, which can be viewed as a  ‘linearization’ of the state space, allowing for efficient linear modeling of the time series.
>
> > (2) The benchmark dataset created in the article consists of two parts: SYNTHETIC and REAL-LIFE. However, while the REAL-LIFE part mainly comprises existing time-series datasets, the contribution of the SYNTHETIC part in addressing the limitations of the existing REAL-LIFE datasets is not adequately explained...
>
>
> The synthetic datasets we introduced in the paper complement the typical real-life datasets, resulting in the most diverse and comprehensive LTSF evaluation dataset. Synthetically generated data provide many features not present in typical real-life datasets. Synthetic datasets provide an opportunity for testing, especially long horizons. Contrary to the typical real-life datasets, each trajectory within the synthetic datasets was generated independently, starting with a random initial condition without incurring any overlaps of trajectories.
>
> We emphasize that the synthetic dynamical systems data possess a different set of characteristics than the real-life data, making forecasting difficult in the long term. The features include nonlinear dependencies between different variables and chaotic dynamics (high dependence on the initial state). The dynamical systems that we employed are modeling diverse physical processes, including the weather (the Lorenz eq.), a biological model (the Mackey-Glass eq.), flame propagation (the Kuramoto-Sivashinsky eq.), phase separation of two components of a binary fluid (the Cahn-Hillard eq.), and sophisticated robotic simulators based on rigid-body models (MuJoCo datasets).
>
> > Regarding experiments:
>
> > (1) The description of the process for jointly evaluating models using artificial and real-life data is overly concise and fails to analyze the fundamental differences between artificial and real-life data. It is recommended to propose a systematic evaluation framework...
>
> We agree that a framework that provides an in-depth analysis of the topic is needed, though we think that providing the datasets in question, outlining the issue with modern models, and proposing improvements to the baseline models (DeepAR + CL and Latent N-Linear) for the task provides a good ground for future research in the field.
>
> > (2) The experimental evaluation of Transformer models only includes one model, PatchTST, thereby lacking a comprehensive exploration of other Transformer models. It is advisable to include a wider range of Transformer models in the experiments for a more comprehensive comparison and evaluation.
>
> Thank you for suggesting exploring other Transformer models. We will include the Informer model, which is probably the most popular LTSF Transformer, in a future revision of the paper.
>
> > Regarding presentation:
>
>  > (1) The "BENCHMARK SYNERGY" section lacks formulas and illustrations, making it difficult for readers to understand its content. It is recommended to provide more visualizations and illustrations in this section to aid reader comprehension.
>
> The “Benchmark Synergy” section aims to discuss how the real-life and synthetic datasets complement each other within the Unified LTSF benchmark. Regarding the introduced synthetic datasets, the detailed datasets presentation and the generating dynamical systems formulas are provided in the Section before (Sec. 2.  Datasets).

---

> ### Author Response · Authors · 2023-11-21
> **Answer to Reviewer ptT1 part 2.**
>
> > (2) The article lacks basic visual analysis and conclusions, making it challenging for readers to grasp the experimental results intuitively. Including more data visualizations and clear conclusions in the article would provide more intuitive information.
>
>
> Regarding the basic conclusions from the benchmark, we note that the paper contains the whole section summarizing the main conclusions from our study - see Section 5. Benchmark Conclusions. Nonetheless, we reiterate the four main conclusions from the benchmark below:
> 1. **Need of sanity check datasets**. We demonstrate that established models like DeepAR and LSTM fail to forecast simple signal structures, like the sum of two sinusoidal waves, and exhibit convergence issues even with extensive lookback windows in such simple cases.
> 2. **Best models depend on the dataset**. The notion that newer models outperform older ones is challenged by our findings. While the newer models (N-Hits, NLinear, SpaceTime, PatchTST) dominate the older LSTM and DeepAR approaches only in univariate real-life data (see Tab.4), the situation changes dramatically with multivariate and synthetic datasets. This is important as the new methods are often evaluated using solely the established univariate real-life datasets, entailing the risk of overfitting to datasets with specific characteristics and avoiding the question about the methods' generalizability.
> 3. **Underappreciated baselines: Classical NN models**. We emphasize that the classical approaches, LSTM and DeepAR, have often been overlooked as baselines. However, our experiments reveal their consistently strong performance compared to state-of-the-art models.
>
> 4. **Our custom DeepAR + CL and Latent LTSF models are competitive**. We emphasize that DeepAR + CL and Latent LTSF models beat their vanilla counterparts in almost the entire benchmark, which makes them legitimate candidates for TS forecasting baselines.
> Regarding the basic visual analysis, we are unsure what the reviewer means by ‘visual analysis’. Please note that most of the used datasets are multivariate (including up to 325 variables), and we benchmark nine different models in total, making the visualizations hard to present comprehensively, especially given the conference paper styling. Instead, we rely on tables summarizing achieved metrics (MAE & MSE) by the methods for all tested datasets. In this regard, we follow the established methodology in the field.
>
> >  (3) The article lacks intuitive descriptions of the datasets, such as data-time graph representations, making it difficult for readers to understand the characteristics and structure of the datasets clearly.
>
>
> Given the multitude of datasets and often a large number of variables in the dataset (up to 325), we are not convinced that data-time graph representations can be concisely and at the same time comprehensively presented, given the conference paper content length restrictions.
> Instead, to present the dataset's structure and demonstrate that each variable found in the dataset is generally sound and non-trivial (e.g., constant), we rely on the basic statistical description of each individual variable in the dataset; such statistics are presented in Tables 12-16 in the Appendix. For the case of the multi-variate datasets, we also analyze the Pearson correlation coefficient matrices, presented in Fig. 2 in the Appendix.
>
> > (4) There is a lack of illustrations for different data generation methods, impeding readers' understanding of the data generation process and methods. It would be beneficial to include relevant figures ...
>
>
> Again, we are unsure what the reviewer means by the illustrations of the data generation methods in this context. Data-generating methods are based on either invoking an established sophisticated simulator in the robotics domain (MuJoCo) or employing a standard method of applying a numerical scheme discretizing in time (and in space in the case of PDEs) for solving a continuous dynamical system. The details of the dynamical system and the numerical method for each introduced synthetic dataset are provided in Section 2 of the paper. We find such a description precise enough for the eventual reproduction of results by an informed reader. Please refer to the provided references for a more detailed description of the dynamic process behind the model and visualizations of the flows. The dynamical systems that we employed model various physical processes, including the weather (the Lorenz eq.), a biological model (the Mackey-Glass eq.), flame propagation (the Kuramoto-Sivashinsky eq.), phase separation of two components of a binary fluid (the Cahn-Hillard eq.)

---

> ### Author Response · Authors · 2023-11-21
> **Answer  to Reviewer ptT1 part 3.**
>
> > (5) The absence of data-time graph representations for the predictive performance of different models hinders readers' ability to visually compare the performance of various models. It is recommended to provide data-time graph representations in the results section to facilitate a better understanding of the predictive performance of the different models.
>
>
> Please note that most of the used datasets are multivariate (including up to 325	variables), and we benchmark nine different models in total, making the visualizations of the data-time performance of models hard to present concisely and comprehensively.
> We believe that the predictive performance is most accurately described by tables summarizing achieved metrics (MAE & MSE) by the methods for all tested datasets; this form of comparing LTSF methods aligns with existing work on ML models for LTSF.

---

### Official Review · Reviewer_WuTL · 2023-11-01

**Soundness:** 1 poor
**Presentation:** 2 fair
**Contribution:** 1 poor
**Rating:** 5
**Confidence:** 2

**Summary:**

**Summary:**

This paper presents a comprehensive dataset specifically designed for long-term time-series forecasting, which includes simulated data as well as real-life data. The authors standardize each dataset into training and test trajectories with predetermined lookback lengths and conduct an extensive benchmarking analysis using both classical and state-of-the-art models, including a custom latent NLinear model and an enhanced DeepAR with a curriculum learning phase.

**Strengths:**

1. Several simulated datasets are proposed.

2. lookback windows are standardized.


**Weaknesses:**

1. I'm not fully convinced that including various simulated datasets would be helpful. One significant feature of long-term forecasting is high volatility, such as weather and stock prices. The evolving procedure can hardly be described by several relatively simple equations. Thus even if a model works well in the simulated dataset, it still may not necessarily also work well in real-world datasets.

2. No new real-world datasets are proposed.


**Questions:**

1. I'm wondering if the authors could elaborate more on the necessity of including simulated datasets and the performance correlation between simulated datasets and real-world datasets.


At the current stage, the paper's contributions, while noteworthy, do not seem to meet the high threshold of a top-tier machine learning conference like ICLR.  However, I'm not a expert in dataset track and I am open to reconsidering my decision after rebuttal .

**Strengths:**

Please refer to the Strengths section in Summary.

**Weaknesses:**

Please refer to the Weaknesses section in Summary.

**Questions:**

Please refer to the Questions section in Summary.

---

> ### Author Response · Authors · 2023-11-21
> **Answer to Reviewer WuTL**
>
> We thank the reviewer for the insightful remarks. We address the raised concerns and questions below.
>
>
> > No new real-world datasets are proposed.
>
> Our goal in the paper is to provide a unified LTSF benchmark, including now-standard univariate real-life LTSF datasets, supplemented with spatio-temporal traffic dataset with a large number of variables (PEMS-Bay) and the proposed new synthetic datasets. We do not provide new real-life datasets but rely on existing sources, as a few standard univariate real-life datasets are picked from the established Monash time series archive. Whereas multivariate real-life data is more scarcely available than univariate, we propose to employ an instance from the existing multivariate traffic data (Pems-Bay). On top of that, synthetically generated data provides an opportunity for evaluating the ML forecasting models in further complementary scenarios, with data bringing in different characteristics and challenges in forecasting.
>
> > I'm not fully convinced that including various simulated datasets would be helpful. One significant feature of long-term forecasting is high volatility, such as weather and stock prices. The evolving procedure can hardly be described by several relatively simple equations. Thus even if a model works well in the simulated dataset, it still may not necessarily also work well in real-world datasets.
>
> > I'm wondering if the authors could elaborate more on the necessity of including simulated datasets and the performance correlation between simulated datasets and real-world datasets.
>
> We concur with the reviewer in the belief that high volatility is a significant feature of long-term forecasting, and therefore, we included in our benchmark some of the typically employed univariate real-life time-series datasets: electric, M4, ETT. However, the belief is that the newer models will perform better in diverse LTSF scenarios, including datasets characterized by a lesser degree of volatility, like the synthetic data generated from simulators or by directly solving dynamical systems.  We emphasize that the synthetic dynamical systems data possess different characteristics, making forecasting difficult in the long term, including nonlinear dependencies between different variables and chaotic dynamics (high dependence on the initial state). The dynamical systems that we employed are modeling various physical processes, including the weather (the Lorenz eq.), a biological model (the Mackey-Glass eq.), flame propagation (the Kuramoto-Sivashinsky eq.), phase separation of two components of a binary fluid (the Cahn-Hillard eq.), and robotic models (MuJoCo datasets).
>
> Moreover, synthetically generated data provide many features not present in typical real-life datasets. Synthetic datasets provide an opportunity for testing, especially long horizons. Contrary to the typical real-life datasets, each trajectory within the synthetic datasets was generated independently, starting with a random initial condition without incurring any overlaps of trajectories.
> Please note that our extensive benchmark debunks the belief that the newer models will perform better in diverse LTSF scenarios, demonstrating that newer state-of-the-art ML models (NHITS, NLinear, PatchTST, Space-Time)  outperform classical methods in real-world datasets but classical approaches (DeepAR & LSTM) outperform the newer models in the case of synthetic datasets.
>
> We also find that established models like DeepAR and LSTM fail to forecast simple signal structures, like the sum of two sinusoidal waves, and exhibit convergence issues even with extensive lookback windows in such simple cases.

---

### Official Review · Reviewer_av6e · 2023-11-07

**Soundness:** 2 fair
**Presentation:** 2 fair
**Contribution:** 2 fair
**Rating:** 3
**Confidence:** 4

**Summary:**

The authors propose a standardised time series dataset for use in benchmarking long-term time series forecasting (LTSF) methods. A variety of methods are tested on the dataset, with slightly simpler methods (NLinear and DeepAR-CL) demonstrating consistently better performance.

**Strengths:**

Providing researchers with additional datasets for testing would help to strengthen the claims of new LSTF architectures proposed.

**Weaknesses:**

However, it is not immediately clear what novel methods the authors have created and what the value proposition of the paper is. While additional datasets can be beneficial to strengthen claims, it is not immediately unclear why the existing datasets for benchmarking (often real world diverse datasets) are insufficient, or why the synthetic datasets proposed (which can also be found in other time series papers, particularly MuJoCo) are better models the LSTF problem. In addition, details on hyperparameter tuning are sparse, with critical hyperparams such as learning rates and regularisation params (e.g. dropout) omitted from the paper. Given the diversity of time series datasets, performance of hyperparams are highly dataset specific -- and without full tuning it is difficult to disenteagle if underperformance is due to improperly selected hyperparams (e.g. with LSTM on sine waves). This is particularly the case for larger transformer models, especially when transferred onto simpler datasets.

**Questions:**

1. How would authors describe the novelty of the paper, and what new methods have been created/proposed?
2. Why are the synthetic datasets suggested better suited for the LSTF problem?
3. How is hyperparam tuning concretely performed, and how are learning rates/regularisation params set?

---

> ### Author Response · Authors · 2023-11-21
> **Answer to Reviewer av6e**
>
> We thank the reviewer for the remarks. We address the raised concerns and questions below.
>
> > How would authors describe the novelty of the paper, and what new methods have been created/proposed?
>
> The main novelties of the paper are summarized in the final part of the introduction. We reiterate the main novelties of our paper below:
>
> * We aim to provide a more representative benchmark dedicated to LTSF. Apart from the typical real-life datasets (also included in our benchmark), we create a series of synthetically generated datasets employing diverse methodologies presented in the paper. Notably, the synthetic datasets provide an opportunity for testing especially long horizons, and each trajectory within the synthetic datasets, contrary to the typical real-life datasets, was generated independently, starting with a random initial condition without incurring any overlaps of trajectories. The train and test sets have separate trajectories, avoiding potential issues with train/evaluation data overlaps, etc.. Overall, we created over 100 GB of data, resulting in the largest LTSF dataset available.
>
> * We comprehensively evaluate the existing LTSF ML approaches on the created unified LTSF benchmark. We include all recent state-of-the-art LTSF approaches, supplemented by the classical methods (DeepAR & LTSM). The belief is that the recently introduced methods will perform better in diverse LTSF scenarios. In our benchmark, we debunk this belief. The newer methods perform better in the typical real-life scenarios; however are being outperformed in the synthetic datasets scenarios, raising many question marks and requiring further investigation beyond the current paper.
>
> * We introduce two new model architectures - a custom latent NLinear model and an enhancement of DeepAR with the curriculum learning phase. We observed significant improvements across the entire dataset when using these models, suggesting their effectiveness as baselines for LTSF.
>
> * We demonstrate that established models like DeepAR and LSTM fail to forecast simple signal structures, like the sum of two sinusoidal waves, and exhibit convergence issues even with extensive lookback windows in such simple cases.
>
> > Why are the synthetic datasets suggested better suited for the LSTF problem?
>
> We do not claim that the synthetic datasets are better suited for the LTSF problem. We rather claim that the synthetic datasets we introduced in the paper complement the existing typical real-life datasets, resulting in the most diverse and comprehensive LTSF evaluation dataset up to date. Synthetically generated data provide many features not present in typical real-life datasets. Synthetic datasets provide an opportunity for testing, especially long horizons. Contrary to the typical real-life datasets, each trajectory within the synthetic datasets was generated independently, starting with a random initial condition without incurring any overlaps of trajectories.
>  As described above, we comprehensively evaluated ML approaches with some surprising conclusions.
>
> > How is hyperparam tuning concretely performed, and how are learning rates/regularisation params set?
>
> We have tuned the most important hyperparameters of the two introduced models (Latent N-Linear \& DeepAR + CL). In the case of Latent N-Linear, we varied the encoder width and the latent space dimension. The latent N-Linear model is pretty robust w.r.t. its hyperparameters. In the case of DeepAR + CL, we varied the model depth and the hidden dimension. The results for DeepAR + CL are generally within the std. dev. margin for the smaller architectures and degrade when using larger architectures (hidden dim $>128$ or depth $>3$). We provide an ablation study in the paper appendix (Table 11). Regarding adjusting the learning rate, for a fair comparison, we used the same learning rate (0.001) of the Adam optimizer (this is, in fact, the for all tested nonlinear ML models and a smaller learning rate (0.0001) for the NLinear model (vanilla & latent) due to significantly faster convergence of linear vs nonlinear models.

---

### Meta-Review · Area_Chair_WkBg · 2023-12-07

**Metareview:**

The paper proposes a new time series dataset collection for benchmarking for Long Time Series Forecasting, which consists of synthetic and real world datasets. The authors evaluate state-of-the-art forecasting approaches along with variants of simpler architectures on this benchmark.

The contributions in the paper unfortunately fall below the bar for the conference. While the community would definitely be well served with a larger and more comprehensive dataset for LTSF benchmarking, reviewers pointed out several concerns with this dataset. Firstly a large portion of the dataset is synthetic, and the value derived from synthetic datasets isnt well motivated here, qualitatively or quantitatively.
On the other hand, most of the real world datasets in this collection are not new and have been used previously in the community. I would urge the authors to work on adding a more comprehensive set of real-world datasets to this collection, and/or to better motivate the need for synthetic datasets.

**Justification For Why Not Higher Score:**

The contributions in this paper do not meet the conference bar, since the value of this collection of datasets is not clear - the real-world datasets here are not new, and it is not clear how useful the synthetic datasets would be to researchers in this field.

**Justification For Why Not Lower Score:**

N/A

---

### Decision · Program_Chairs · 2024-01-16

Reject